# Early detection of depression using a conversational AI bot: A non-clinical trial

**Payam Kaywan⦿, Khandakar Ahmed⦿ ⓘ◉\*, Ayman Ibaida, Yuan Miao, Bruce Gu**

Intelligent Technology Innovation Lab, Victoria University, Melbourne, Victoria, Australia

◉ These authors contributed equally to this work.
\* khandakar.ahmed@vu.edu.au

**Data Availability Statement:** All data files are within the following GitHub repository: https://github.com/entenam/Depression-Analysis-DEPRA-chatbot.

## Abstract

### Background

Artificial intelligence (AI) has gained momentum in behavioural health interventions in recent years. However, a limited number of studies use or apply such methodologies in the early detection of depression. A large population needing psychological—intervention is left unidentified due to barriers such as cost, location, stigma and a global shortage of health workers. Therefore, it is essential to develop a mass screening integrative approach that can identify people with depression at its early stage to avoid a potential crisis.

### Objectives

This study aims to understand the feasibility and efficacy of using AI-enabled chatbots in the early detection of depression.

### Methods

We use Dialogflow as a conversation interface to build a Depression Analysisn (DEPRA) chatbot. A structured and authoritative early detection depression interview guide, which contains 27 questions combining the structured interview guide for the Hamilton Depression Scale (SIGH-D) and the inventory of depressive symptomatology (IDS-C), underpins the design of the conversation flow. To attain better accuracy and a wide variety of responses, we train Dialogflow with the utterances collected from a focus group of 10 people. The occupation of the focus group members included academics and HDR candidates who are conscious, vigilant and have a clear understanding of the questions. In addition, DEPRA is integrated with a social media platform to provide practical access to all the participants. For the non-clinical trial, we recruited 50 participants aged between 18 and 80 from across Australia. To evaluate the practicability and performance of DEPRA, we also asked participants to submit a user satisfaction survey at the end of the conversation.

### Results

A sample of 50 participants, with an average age of 34.7 years, completed this non-clinical trial. More than half of the participants (54%) are male and the major ethnicities are Asian (63%), Middle Eastern (25%), and others 12%. The first group comprises professional

**Funding:** The author(s) received no specific funding for this work.

**Competing interests:** NO authors have competing interests.

academic staff and HDR candidates, the second and third groups comprise relatives, friends, and volunteers who were recruited via social media promotions. DEPRA uses two scientific scoring systems, QIDS-SR and IDS-SR to verify the results of early depression detection. As the results indicate, both scoring systems return a similar outcome with slight variations for different depression levels. According to IDS-SR, 30% of participants were healthy, 14% mild, 22% moderate, 14% severe, and 20% very severe. QIDS-SR suggests 32% were healthy, 18% mild, 10% moderate, 18% severe, and 22% very severe. Furthermore, the overall satisfaction rate of using DEPRA was 79% indicating that the participants had a high rate of user satisfaction and engagement.

## Conclusion

DEPRA shows promises as a feasible option for developing a mass screening integrated approach for early detection of depression. Although the chatbot is not intended to replace the functionality of mental health professionals, it does show promise as a means of assisting with automation and concealed communication with verified scoring systems.

## Introduction

Approximately 264 million people worldwide [1] and 3 million Australians [2] suffer from mental disorders such as anxiety or depression. Depression is a primary cause of disability, putting a strain on the global illness burden. In contrast to normal emotions, a depressed individual has persistent symptoms and a loss of interest in life [3]. The consequences might be more severe, impacting a variety of social settings, such as poor performance at school, job, or family relationships. The physical and behavioural symptoms include changes in sleep, appetite, energy level, concentration, daily behaviour, and self-esteem [4]. Inappropriate depression treatment might potentially result in suicide. The study reveals that around 80% of mental health patients go untreated when adequate therapy is not accessible [5]. Inadequate resources, a shortage of skilled mental healthcare personnel, and societal stigma contribute to the ineffective treatment of mental health concerns.

Computer-assisted mental health (CAMH) solutions offer mental health information and services such as psychotherapy, behavioural treatment, early depression detection and intervention that are evolving and becoming an integral part of the health system [6]. The emergence of artificial intelligence (AI) is broadening the scope and possible use of CAMH solutions [7]. CAMH solutions are economically sustainable, effective and complementary to conventional mental health care. They have the capability and broad availability to extend to marginalised populations, alleviate economic burdens, and assist in avoiding stigma [6].

A wide range of computer-assisted therapy solutions has been developed over the past few years to support the treatment of mental disorders and behavioural health problems [8]. Variations include but are not limited to conversational chatbots, mobile and web-based applications [9] facilitating real-time symptoms tracking, medication management, early detection, psychological intervention [10], personalized monitoring and recovery plans, lifestyle improvement, or acting as a digital therapist [11].

A widely used CAMH application is the chatbot, also known as a conversational agent [12]. A chatbot is a computer application that replicates and processes human dialogue. It allows humans to communicate with a digital assistant as if conversing with an actual human [13]. A

data-driven chatbot, also referred to as a virtual assistant is more interactive, sophisticated and customised. These chatbots are context-aware and use natural-language understanding (NLU) [14], natural language processing (NLP) [15], and machine learning [16] to learn the user inputs and generate output. In the longer term, a chatbot can employ analytics and predictive intelligence to provide personalization based on user profiles and previous activity. Over time, digital assistants may learn user preferences, provide recommendations, and foresee requirements. They can start dialogues in addition to monitoring data and intent.

CAMH is a highly desirable support service due to several factors [17]. The preliminary data indicate that participants or patients prefer to communicate with a non-human conversational bot when sharing specifics about their circumstances [18]. Another study [19] reveals that 70% of users prefer mobile health applications (mHealth) over face-to-face treatment when used to self-manage or self-monitor. Furthermore, a non-clinical randomised trial platform [20] demonstrates the effectiveness of employing an AI-based computer-assisted cognitive-behavioural therapy (CCBT) to alleviate the self-reported symptoms of depression and anxiety in college students.

In summary, the community is currently open and positive on artificial intelligence and chatbots to help with mental health concerns. However, early detection before the onset of the disease is critical but is often overlooked. With this motivation, we proposed and developed a non-clinical chatbot, DEPRA, with social attributes to help users with mental health concerns and detect the illness in its early stage. This is a preliminary study which aims to provide a simple rather than a complex platform to validate its feasibility and efficacy.

DEPRA is capable of assisting participants examine their mental health conditions as a reference so that medical professionals can assist patients who are suffering from depression.

- It was implemented following a conversation flow based structured early detection depression interview guide for the Hamilton Depression Scale (SIGH-D) [21] and Inventory of Depressive Symptomatology (IDS-C) [22] drawing on the experience of a group of psychiatrists. This makes the DEPRA chatbot a unique chatbot in that it shares the same set of clinically proved questions in the design of its questions for interaction with participants.

- The validity of the questions used in the DEPRA chatbot was measured by applying closed group participation. This was arranged by designing open-ended questions for the participants. DEPRA is integrated with Facebook Messenger so, participants can interact with the chatbot through social media to share their responses.

- The questions asked by the DEPRA chatbot take approximately 30 minutes to complete. As the results suggest, the participants' responses indicate high satisfaction rates. They reported that the questions were easy to comprehend and respond to, it was not as time consuming as attending a face-to-face psychiatrist session, and they preferred the option to send text messages via social media platforms compared to talking to a psychiatrist in a consultation session.

## Related works

Since 2015 [23], chatbots have been developed as digital assessors with two primary goals: a) to be psychological intervention utilities that assist human beings with their mental health, way of life, and proclivity to work within their careers; and b) to assess the depression level of humans. We categorized the related works into two sections, namely works focus on technological-based depression detection and works which focus on chatbots which provide psychological interventions.

## Psychological intervention

As the demand for medical professionals increases, utilizing chatbots to cooperate with and assist in the medical field will help with requests for assistance and ease the demand on medical practitioners. Ly et al. [24] conducted research to measure the effectiveness of smartphone apps in CBT interventions. A total of 28 participants (both males and females) took part in the Shim chatbot intervention which used a smart phone app as a text-only method of collecting data over a two-week period. The research confirmed that the participants' experiences and the output of their conversations with the Shim chatbot [25] can promote mental health. The duration of one face-to-face therapy session was 60 minutes. The results show that there is no significant inconsistency between a blended treatment (which comprised four face-to-face sessions and the smartphone application) and the full behavioural activation (which comprised including ten face-to-face sessions and no smartphone application) on the result variances. Both pre- and post-measurement and follow up actions were considered in this study. Sharma et al. [26] designed a chatbot application for Android devices which was designed as a virtual psychotherapist. Depression levels were measured on a scale of 0 to 4, with 0 being completely healthy and 4 being highly depressed. The strategy behind the question design was that for any optimistic response, the total score will increase by an x greater than zero value. The depression levels were divided into a) zero depression, absolutely healthy b) stressed c) highly stressed d) depressed e) highly depressed. The research results affirmed that it is quite difficult to extend therapy chatbot results because the assessments are not managed constantly.

Gardiner et al. [27] conducted research on a group of urban females to study the changes in their lifestyle by applying an embodied conversational agent (ECA) as the basis of implementation. Gabby [27], a 3D chatbot, divided the participants, who were mostly females, into two categories: a) an ECA that included mental health, stress level, physical exercise, and eating tendencies; b) the same as group (a) with the additional factor of mediation to be monitored every day for the duration of one month. Bickmore et al. [28] designed and implemented a 3D chatbot to inform a patient of their hospital discharge plan. The purpose of this study was to ascertain the attitudes of hospitalised patients who were being prepared by the chatbot to start the discharge procedure. Podrazhansky et al. [29] designed a chatbot to collect data through text, voice, or video. The chatbot mobile app was designed to evaluate the mental health of its participants. Machine learning (ML) algorithms along with natural language processing (NLP) techniques were applied to detect mental disorder, and in this instance, prevention orders were offered to the participants. Sharma et al. [30] developed a chatbot which was capable of stress prediction and management. The study reflects on the serious physical and mental diseases that can result as a consequence of long-term stress, such as heart attacks and depression. Tielman et al. [31] developed a system, 3MR-2, which uses a digital diary and a 3D World-Builder. The system is used to recreate traumatic moments in the life of participants to help them recover from Post-Traumatic Stress Disorder (PTSD). The WorldBuilder chatbot has several limitations and the drawbacks of not being scalable and being overly reliant on the programmer. Unless this chatbot agrees to the existence of each object, there are no matches.

Lucas et al. [32] introduced the concept of a 3D chatbot as a digital assessor to consult with military veterans who were suffering from severe symptoms of PTSD. A study by Shinozaki et al. [33], developed a conversational agent to assist IT professionals suffering from anxiety and stress due to low success rate of IT projects.

Woebot [34] is an automatic conversational chatbot based on CBT, implemented on Facebook Messenger. Experiments were conducted to evaluate Woebot using college students as participants. The results indicated that participants' depressive symptoms dramatically decreased. The students commented that their interactions with Woebot were more enjoyable

than the therapy sessions held by health care professionals. However, Woebot does not provide a comprehensive process of CBT and it mainly deals with psychoeducation for stress control. Participants need to maintain their level of enthusiasm to interact with Woebot for a period of time before the final results are achieved [35].

In an unblinded trial conducted by Fitzpatrick et al. [36], 70 participants from a university community social media site in the age range of 18 to 28 years old were recruited online to evaluate the acceptance rate of college students who reported feelings of anxiety and depression to engage with a conversational agent and participate in a self-help program. The average age of the participants was 22.2 years old, 47 out of 70 were female, 54 out of 58 were non-Hispanic and 46 out of 58 were Caucasian. The participants were randomized to receive either 2 weeks or up to 20 sessions of self-help content derived from CBT principals in a conversational format with the text-based conversational agent (Woebot) (n = 34) or were directed to the National Institute of Mental Health (NIMH) e-book, "Depression in College Students" as an informative control group (n = 36). At the next stage, all the participants were encouraged to complete the web-based versions of PHQ-9, GAD-7 and Positive and Negative Affect Scale at baseline and two-to-three weeks later. [36].

Fulmer et al. [20] conducted a study to assess the feasibility and efficacy of a psychological AI chatbot (Tess) to reduce self-identified symptoms of depression and anxiety in college students. Tess, designed by X2AI, can deliver integrative mental health support, psychoeducation and reminders through a brief conversation (text messages). Tess can react to the participants' emotional needs by analysing the conversational content through textual interaction or by Facebook Messenger. In the research, the researchers conducted a randomised controlled trial over 74 participants recruited from 15 universities across the united states. Two test groups (n = 26 & n = 24) received unlimited access to Tess for 2 and 4 weeks, respectively, whereas the third controlled group (n = 24) received limited access to Tess only after completion of an online study following a given link of National Institute of Mental Health's (NIMH) eBook on depression among college students. The survey used Patient Health Questionnaire (PHQ-9), Generalised Anxiety Disorder Scale (GAD-7), and Positive and Negative Affect Scale (PANAS) to measure depression symptoms, anxiety disorder and satisfaction levels. The study found that both test groups experienced a significant reduction in anxiety and depression symptoms, and their satisfaction was higher than that of the control group. The study reveals that AI can offer cost-effective integrative psychological solutions complementary to traditional treatment in reducing symptoms of depression and anxiety.

The aforementioned chatbots focus on psychological intervention with depression detection. However, as stated in the previous section, identifying depression in its earliest stage can minimize the impact on public health by potentially reducing the escalation of the disease. None of the above intelligent conversational chatbots focus on early depression detection with a feasible and efficient platform.

## Early depression detection

A study by Kharel et al. [37] researched the early detection of depression with the aid of machine learning that was performed on patients' MRI images. The Functional Magnetic Resonance Imaging (fMRI) dataset consists of the brain images of 30 participants who were on antidepressants or CBT. Due to limitations results from the small size of the dataset, the use of ML algorithms and features were restricted. The study found that Deep Learning algorithms and in particular Convolutional Neural Networks (CNN) were the most popular methodology to analyse medical images. Also, Support Vector Machines (SVM) were utilized to provide a measure and value of accuracy. The classification method of facial processing and developing

depression neurobiological patterns were the initial purpose of this study. The cerebral formation over the entire surface of the brain in healthy participants was examined. The idea behind this was to predict neurocognitive disorder in a healthy population. In the data collection phase, 38 participants were recruited. Half of the individuals (N = 19) were suffering from depression; however, they were not on any medications, the other half (N = 19) were healthy and willing to participate in the study. The fMRI data were utilized at each stage of the process with the aid of an SVM pattern. As a result, the work in the area of the early detection of depression remains an intact base of knowledge that requires further work and study. Kharel et al. utilized methods to collect data which are not currently applied in other studies such as audio, video and even movement.

Philip et al. [38] conducted research with 179 participants for a period of 1 day using an Embodied Conversational Agent (ECA). ECA is a form of intelligent user interface (AVATAR) that combines voice, facial and gesture expression to conduct face-to-face interview. ECAs have high potential but they have not been used often in studies on mental disorders. A total of 90 participants underwent the medical interview with the psychiatrist before the session with ECA and the remaining 89 participants were visited by the psychiatrist after the interview with the ECA. The average age of the participants was 46.5 years, 57.5 percent of the participants were female and the average number of years of education undertaken by the participants was 13.3 years. A total of 35 participants (19.6 percent) were diagnosed with Major Depressive Disorder (MDD) confirmed by a psychiatrist. For the group that was examined by the psychiatrist before the ECA session, the average age was 45.7 years, females were 55.6 percent of the participants and the average number of years of education undertaken by the participants was 13.1 years. In the group which was examined by the psychiatrist after the ECA session, the average age was 47.3 years old, females formed 59.6 present of the participants and the average educational level was 13.1. As indicated by this information, the two groups were almost identical and there were no major differences between them. The decision algorithm implemented in ECA correctly identified 17 out of 35 patients diagnosed with MDD by the psychiatrist, indicating enhancement needed in the diagnostic algorithm. However, the study had two goals: a) to test the efficiency of diagnostic systems related to MDDs; b) to evaluate the acceptability of using an ECA. On a scale of 0–30, 73% of patients score ECA above 24. The overall acceptability score of ECA by participants was 25.4± 4.6. The conclusion of this study suggests that ECAs have a high potential to be utilized for practical and standard applications in interviews.

Our proposed DEPRA chatbot falls under this bot family category as a depression detection bot, the aim of which is to circumvent some of the existing challenges that have been discussed in this section. The main difference between this research and the related works conducted so far is that the DEPRA chatbot gathers data based on a series of psychologically approved questions and its purpose is to triage and detect the early signs of depression. However, the available chatbots mostly focus on the depression disorder itself and try to find a better cure. In other words, the DEPRA chatbot aims to assist medical professionals detect and cure depression in its early stage and it does not aim to replace a medical professional. DEPRA was implemented to further investigate the application of AI to interact with users, collect their responses to questions on depression symptoms and identify depression and estimate the level of its severity.

## Methodologies

The objective of this study is to consider the efficacy and feasibility of applying an AI-based chatbot for early depression detection. DEPRA follows the textual conversations and interactions between the participants and the agent. Early depression detection is the core focus of

this research. We are aware that the early detection of depression presents significant possibilities for treating patients and providing support to vulnerable members of society. DEPRA eliminates dependence on a limited set of multiple-choice replies in favour of open-ended responses, allowing participants to express their thoughts and sentiments spontaneously and without hesitation. DEPRA assesses mood, guilt, suicidal tendency, insomnia, agitation or retardation, anxiety, weight change, and bodily symptoms to diagnose depression levels.

To determine the efficacy of utilising a chatbot for mass depression screening, we recruited 50 Australian residents. We hypothesise that individuals will feel at ease and have no qualms about interacting with an AI chatbot rather than an actual therapist. We integrated DEPRA into a Facebook page and enabled participants to contribute at their own pace and convenience. We predict that the absence of a booking requirement and the flexibility of participating on their own schedule will eliminate any impediment to obtaining preparatory help and avoiding social embarrassment. Furthermore, we assume that substituting open-ended replies for multiple choice enables individuals to avoid conscious reactions and helps us to capture a participant's genuine situation. To examine the postulations, we directed participants to complete a concise user experience survey after the main session using the same platform. At the completion of the participation, DEPRA scored the replies and classified the participants into five groups based on their SIGH-D and IDS-C scores: nil, mild, moderate, severe and very severe depression.

## Architecture of DEPRA

The trial in this study uses DEPRA as a conversation platform for interaction with the participants. Data collection includes three phases a) digital signing of the consent form, b) questionnaire conversation session, and c) user experience feedback form. DEPRA consolidates the three phases into a single point of interaction. When the participants trigger DEPRA through a greeting message, it first checks the consent status. If needed, DEPRA directs the users to a consent form and asks for a digital signature with official agreement supported by Victoria University, Melbourne, Australia. The participants then return to the main session where they participate in a conversation with 27 questions based on the Hamilton Interview Guideline (HIG) [21]. Upon completion, DEPRA prompts the users to share their experiences through the user experience rating form.

Fig 1 presents the schematic diagram of DEPRA which visualises the implementation and design of the DEPRA chatbot. DEPRA is implemented on the Google Dialogflow platform using Node.js in the Fulfillment Inline Editor. Dialogflow is a Natural-Language Understanding (NLU) platform used to develop a conversational user interface. At the front-end, we integrated DEPRA with the Facebook Messenger application (Fig 2a) to communicate with the

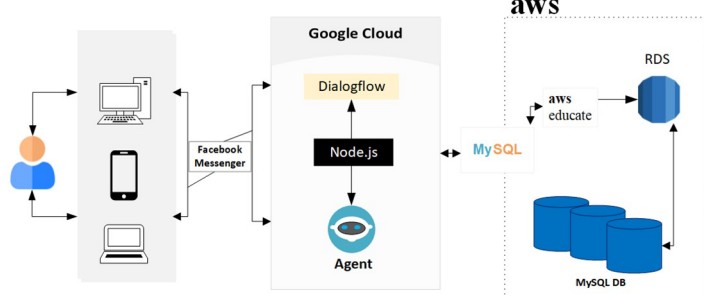

**Fig 1. Schematic diagram of DEPRA—Design of the implementation.**

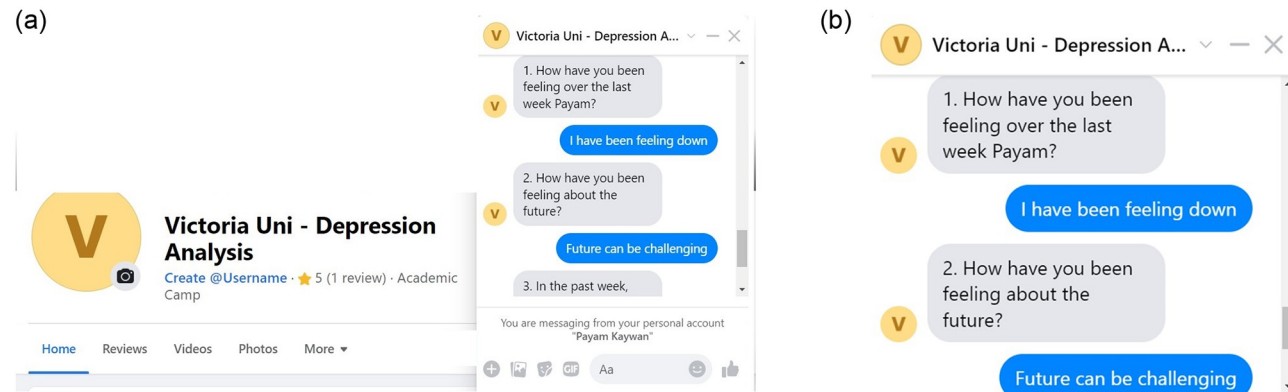

**Fig 2. DEPRA in FB messenger and sample conversation with a participant.** (**a**) Integration with Facebook Messenger. (**b**) Sample conversation between DEPRA and a participant.

end-users. Facebook Messenger provides a convenient environment for participation, it is easy to access the DEPRA page, and the interaction window is designed internally by Facebook. Facebook Messenger also provides a trustworthy platform for users where various data security and privacy techniques are used, such as AI to identify unusual behavioural patterns correlated with harmful activities, reduce the spread of viral misinformation and harmful content, detecting impersonators, and enhancing safeguards for users under 18. Furthermore, the platform also provides end-to-end encryptions which reinforce safety and security and have increasingly become the standard expectation of users for their preferred communications platforms.

Facebook Messenger is a web-based and mobile application-based platform that acts as the proxy between the user and the DEPRA chatbot. At the back-end, fulfilment acts as the backbone of the conversation that uses API to connect to the database which is responsible for storing the conversation and retrieving information for further studies. We hosted our remote central database in the AWS cloud and used the Amazon Relational Database Service (RDS). We used MySQL Workbench for data modelling, SQL development and administration.

In designing the conversation in Dialogflow, DEPRA defines one intent for each question, where the combined intents handle the complete conversation and enables fulfillment functions. Intents comprise training phrases, entities, action, parameters, responses, contexts and events. Each intent passes the parameters through the input and output contexts. To have a meaningful conversation, following the exact sequence as per HIG is imperative and the response to one question is relevant to the context of the answer from the previous question. Therefore, we used the input and output context for each intent and define the next question in the response section to control the flow of the conversation. Fig 2b shows a sample transcript of a participant interacting with DEPRA. To provide a dynamic response, we validate the response received from the user, store it in the central database (Fig 3 shows the schema/ E-R diagram of the central database which includes all the tables in DB and the way they are interrelated) and define a number of functions in the fulfillment. Table 1 lists the descriptions of each aforementioned technical term.

**Conversation design.** DEPRA uses the structured HIG interview guide, developed by Williams et al., [21] in designing conversation. The base of this guide is one of the most widely used clinician-administered depression scales, namely the Structured Interview Guide for the Hamilton Depression Rating Scale (SIGH-D) [21]. The SIGH-D standardizes the manner of administration and scoring of the scale from the original scaling system, the Hamilton Depression Rating Scale (HDRS) [39, 40]. The original guide identified twenty-one items to measure,

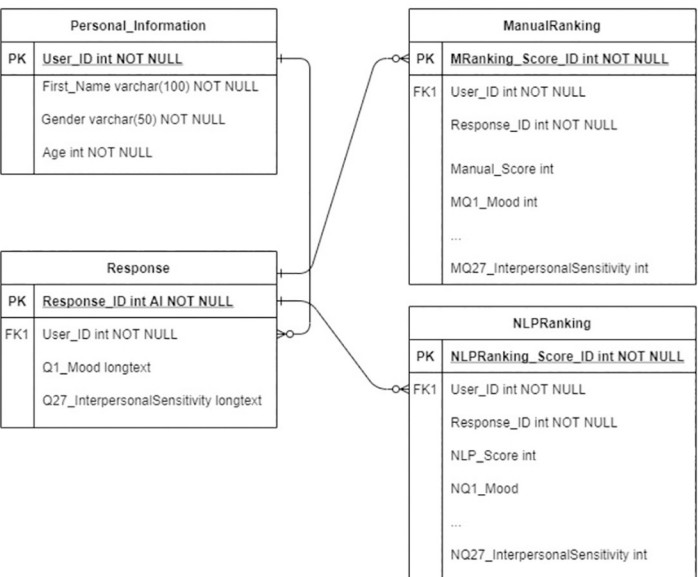

**Fig 3. Schema/E-R diagram of the central database.**

**Table 1. Descriptions of technical terms.**

| Technical Terms | Descriptions |
|---|---|
| Application Programming Interface (API) | Software intermediary allows conversation store into database from Facebook Messenger interactions. |
| Amazon Web Services (AWS) | On-demand cloud computing platforms and providing APIs to our DEPRA. |
| Back-end | This is the "server side" which is responsible for data storage and manipulation and it also communicates with the frontend, sending and receiving information to be displayed as a web page. |
| Chatbot / Conversational Agent | DEPRA application that simulates human-like conversations with users via the Facebook Messenger interface. |
| Dialogflow | Dialogflow is a natural language understanding platform used to design and integrate DEPRA into Facebook Messenger with interactive response systems. |
| Dynamic Response | Responses generated automatically by DEPRA in real-time during the conversation. |
| Facebook Messenger | A social media chat platform, where DEPRA is used for interactions with end users. |
| Front-end | The programming platform for client-side development. |
| Fulfilment Functions | These enable dynamic responses to the particular intent. |
| Intent/s | Intent is what is triggered when users send DEPRA a message. |
| Interface | The chat dialog box offered by DEPRA via Facebook Messenger. |
| MySQL Workbench | A visual database design tool that integrates SQL development, administration, database design, creation and maintenance into a single integrated development environment for the MySQL database system. |
| Natural Language Understanding (NLU) | This involves transforming human language into a machine-readable format for DEPRA. |
| Node.js | A programming language used to create, open, read, write, delete and close files on the server. In addition, it can collect, add, delete, and modify data in our database |
| Relational Database Service (RDS) | RDS is a distributed relational database service offered by AWS. It is a web service running "in the cloud" designed to simplify the setup, operation, and scaling of a relational database for use in DEPRA. |

though Hamilton himself only recommended using the first seventeen since the last four items are loosely associated with the common symptoms of depression. Initially, SIGH-D was designed for hospital in-patients, but later, experts devised many versions. All seventeen items pertain to symptoms that a depressed person could have experience over the past few weeks. SIGH-D was originally unstructured with the least general information for rating individual items. This gap motivates many works focusing on producing structured or semi-structured interview guides. HIG has thirty-one questions to measure seventeen items with 48 sub-questions for further comprehension and each question has four options from which to choose.

In this non-clinical trial, DEPRA uses 27 of the 31 questions and removes multiple choice, sub-questions, and questions with conditional statements. We convert the multiple choice questions into an open-ended response with the help of psychology experts and strictly follow HIG guidelines as well as the SIGH-D and IDS-C scaling system. The rationale behind it is that people generally make conscious choices, and therefore, a group of people tends to exaggerate their condition either positively or negatively. In contrast, this study aims to capture the original state through the participants' descriptions and accept expert assistance to align them to the choices and score them accordingly. Therefore, the participants can interact by explaining their genuine feelings without barriers or distractions to answering openly. Fig 4 presents a summary of the symptoms and their relation to the question flowchart. Appendix A in S1 Appendix provides the complete conversation used in this study.

To implement the conversation in Dialogflow, DEPRA defines a total of 38 intents including SIGH-D questionnaires, welcome messages, personal identifications, confirmation of consent form, database transmissions and other system intents. Table 2 shows a further breakdown of the 38 intents.

## DEPRA training

Training DEPRA based on Dialogflow with adequate utterances for an acceptable and accurate association between intent and response is one of the essential design requirements. Hence, we conducted a closed group survey with ten university staff and students who are the first level of connection to the research. They are aware of the outcomes and methodology of this project (Fig 5 shows the demography of the participants). We use Google Forms as a simplified platform to conduct this closed group survey. Appendix B in S1 Appendix shows the atomic data of all questions from the ten participants. The focus group survey aims to extract a requisite number of keywords to generate utterances of various lengths.

## Participant recruitment

The recruitment of the study participants adhered strictly to Victoria University's ethics guidelines and was approved by the Human Research Ethics Committee. In compliance with these guidelines, all participants must live in Australia during the data collection period, they must be aged between 18 and 80 years and they must be willing to share details of their moods and emotions in a scientific trial. This paper reports the analysis of data from the first 50 participants which was collected until 30th October 2021. The participant recruitment is based on three categories: a) Victoria University staff and HDR Students; b) friends and family members (non-organic group); c) Facebook page (organic group). We used three channels for recruitment (Fig 6)—the first group was university students and university professional staff either full-time, part-time or causal. We emailed a broad group of potential participants using three email accounts with the Webhook link to invite them to participate in this research. We required the academic group to assist in two tasks. Firstly, we recruited 10 participants (N = 10) to share their ideas and responses in a closed group. This was crucial to designing the

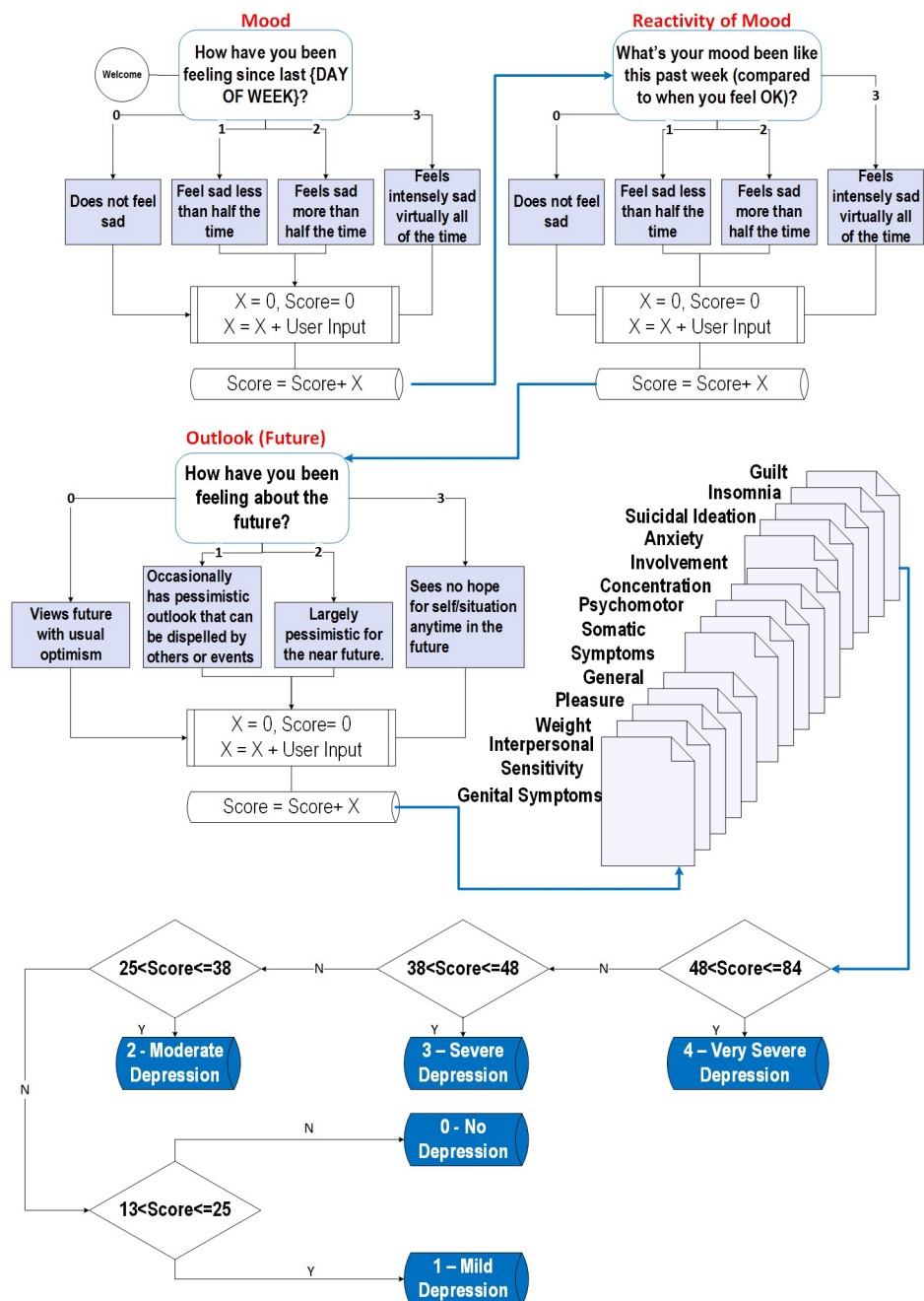

**Fig 4. Flowchart summarizing the symptoms and relations between the questions.**

utterances on Dialogflow as the platform for the DEPRA chatbot. Secondly, we asked the same group to participate in data collection to assist with the research. We then emailed another 65 university staff and HDR students to invite them to participate. We received 16 complete records in our AWS central database. Current academics and HDR candidates were the point of contact as they were presented with the details of the research in weekly and monthly meetings and with these series of updates, they played a critical role in the understanding and comprehension of the requirements of the research. As a consequence, they assisted in responding to

**Table 2. Dialogflow intents breakdown.**

| Type | Number of Intents |
| --- | --- |
| SIGH-D Questionnaires | 27 |
| Welcome, age and gender | 3 |
| Participant's Name/Confirmation of Consent Form | 1 |
| Fallback | 1 |
| Final Intent | 1 |
| Database and MySQL Transactions | 5 |
| TOTAL of Intents | 38 |

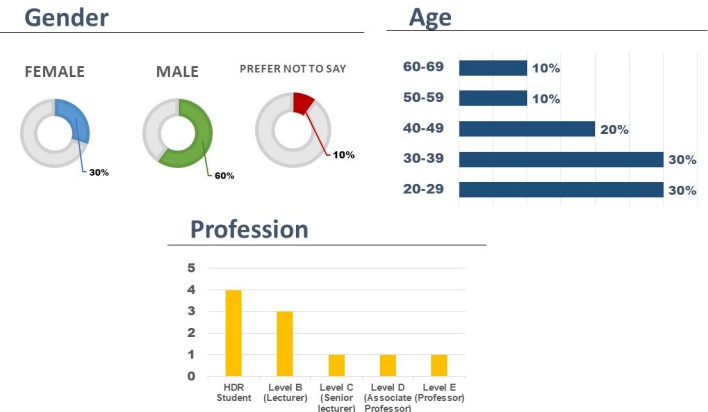

**Fig 5. Statistics of closed group based on gender, age and profession.**

the set of questions and helped to generate the potential responses required for the design of the chatbot.

The second part of the data was collected from friends or relatives who were Australian residents at the time of the data collection. In this phase, we targeted both males and females from various professions, and age groups. We deployed DEPRA on Facebook Messenger and invited 54 participants from social networks—33 signed the consent form but in the end, only

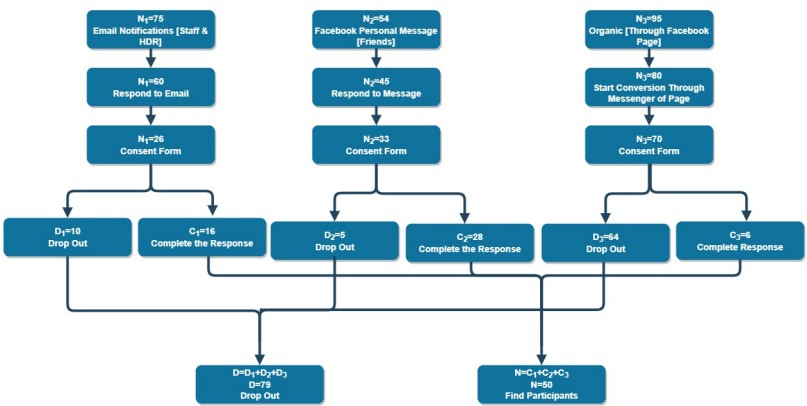

**Fig 6. Participant recruitment flow—3 groups indicating statistics of complete response and dropout rate.**

28 participated in the data collection process. Therefore, we received 28 complete records from the second group. The 44 participants recruited following the first two processes are inorganic, as expected.

The third group, comprising Facebook users keen on sharing their experiences with the DEPRA chatbot, is exciting and organic and is still growing at the time of drafting this manuscript. This group found the page on their own and felt compelled to interact. We registered 8 records from this group, taking the total to 50, excluding duplication and incomplete conversations until the cut-off date. For all three groups, the exclusion criteria include non-residents of Australia, being outside the specified age range, not signing a consent form, and incomplete conversations.

## Ethics and consent form

This research is registered under ethics approval number HRE 20–184 at Victoria University, Melbourne, Australia. It involves a non-clinical population of participants who reside in Australia at the time of the research. The participants are between 18 and 80 years old and willing to reveal information on their moods [41] and emotions [42] in a scientific trial. Appendix C in S1 Appendix provides the consent form for the participants involved in the research, and Appendix D in S1 Appendix provides the Ethics Application related to this study.

The participants indicated their willingness to interact with the DEPRA chatbot and participate in the study by signing the consent form electronically. The participants were prompted to access the consent form after contacting DEPRA. All participants must agree to the terms and conditions in the consent form, which notes that DEPRA may record and analyse their responses to the questions relating to moods, sleep habits, appetite, general health symptoms, etc.

This research deals with human beings and their moods and behaviours in general. The ethics committee follows strict guidelines and conditions to confirm the validity of the methods and approach used by the research team to conduct the research. One of the ethics committee's main concerns is how the study maintains the participants' privacy. After completion of the data collection by the research team, the data is stored on an R drive and a computer with limited access. Only the research team members are allowed access to the raw data. The participants are assured that no unauthorised users will access the raw data. All data were de-identified to protect the privacy of the participants.

Table 3 shows the semantics of the responses w.r.t the extracted keywords, sentiment score and length (min/max) of replies for each question. We use the excerpted keywords to generate one hundred utterances for answers to each question (except those with a numeral), a sentiment score to understand the association between the response and the respondent's state, and length to bring variation within the synthetic utterances.

## User satisfaction form

The set of questions in the user satisfaction form is designed in a way that enables the participants to rate the quality and user experience of the interaction. The questions ask the participant to comment on the ease of understanding and responding to the chatbot, the similarity of attending a psychiatrist session in terms of the time taken, the effectiveness of text messaging in comparison with talking to a psychiatrist, the effectiveness of the sequence of questions to result in a depression level assessment and the likelihood of the participant recommending the interaction to friends and families in Australia. The last question on the satisfaction form is an open-ended question which allows participants to share their comments. All open-ended responses will be used for further training phases for DEPRA.

**Table 3. Sequence of responses.**

| Seq | Data Type | keyword | Sent. Score | Length Min/Max (char) |
|---|---|---|---|---|
| Q1 | String | not bad, good, normal, pumped up, under pressure, not confident, disappointed, down, frustration, challenging | Negative 94.6% | 4/166 |
| Q2 | String | fluctuating, sometimes good, sometimes bad, bit down, excited, opportunities, nervous, a bit anxious, so nervous, all over the place, uncomfortable, down, struggling, out of shape, on the edge, lack of sleep, lot of pressure, fed up, no chance to relax | Positive 58% | 15/125 |
| Q3 | String | hopeful, good, positive, bit frustrated, confident, concentrate on present activities, negative feelings, unwanted challenges, no hopes, difficult times, crisis, more wars, lose home countries, problems, mystery | Positive 75.3% | 4/201 |
| Q4 | String | no, not really, put myself down, judged outcomes, challenged my wife, yes, let a colleague down, down, out of control, satisfied with performance, dedicate to team | Negative 69.7% | 2/232 |
| Q5 | String | lost few relatives, sad, loss, fine, yes, positive, passed away, not feel the same, unfortunately, so many losses, the same feeling, crying, grieving, pressure, lose business, crisis, feeling down, getting worse, passed away, tragic remembrance, lost job, irritated, hopeless, lay off, the same feeling, overwhelmed | Positive 70.5% | 3/259 |
| Q6 | String | not really, no, under control, felt down, down, yes, worse, not bothered with depression, can not breathe, negative thoughts, Could not monitor, depressed | Negative 95.6% | 2/176 |
| Q7 | String | no, too busy, self harm, end my life, yes, meaningless, dead, tough times, never think about self harm, value life, enjoy, very harsh shock | Positive 83.8% | 2/168 |
| Q8 | String | no, yes, loud music, mind so busy, not feel sleepy, no difficulties to sleep, stop crying, normal day, buzzer went off, technical issue, no interruptions | Positive 91.90% | 2/142 |
| Q9 | String | yes, no, woke up to drink water, tough, manage to sleep, could not go back to sleep, scared, nightmare, fix issues | Negative 98.80% | 1/130 |
| Q10 | String | doing nothing, reading novel, watching TV, family, staying with daughter, assisting kids, gardening, discussion, doing laundry, think about a business, studying, house chores, sleeping, listening to music, preparing meals, taking care of nephew, talking to family overseas | Negative 91.90% | 6/118 |
| Q11 | Str/Int | Not Applicable | Neutral 77.20% | N/A |
| Q12 | Str/Int | Not Applicable | Neutral 76.20% | N/A |
| Q13 | String | no, yes, playground, BBQ party, enjoy, essential tasks, No fun, not really, no time for fun, zoom conference call | Negative 58% | 2/130 |
| Q14 | String | poor, not bad, normal, focused, tend to lose concentration, in place at work place, terrible, acceptable level, could not concentrate, focus on studies, act as on-call staff, pretty fine | Negative 91.70% | 4/100 |
| Q15 | String | yes, no, Little bit, slowed down in the morning, get better, deal with skills, slow down in thinking, no problems, up and running | Positive 78.30% | 2/144 |
| Q16 | String | not exactly, little bit, no, yes, on edge, anxious, nervous, perform well, crisis, overwhelmed | Negative 85.60% | 2/128 |
| Q17 | String | no, yes, tense, deal with tasks, irritable, new task, people with corona | Negative 69.30% | 2/101 |
| Q18 | String | no, yes, restless, pressure of life, fidgety, solve problems, normal life, infected patients | pNegative 93.10% | 2/83 |
| Q19 | String | no, not at all, yes, extremely uncomfortable, frightened, uncomfortable, imagined, argue about instructions, number of cases jumped | Negative 95.90% | 2/155 |
| Q20 | String | no, nothing, palpitations, blurred vision, chest pain, increased sweating, hot and cold flashes, dyspnea, no such symptoms | Negative 99.90% | 2/60 |
| Q21 | String | no, constipation, no symptoms, diarrhea, no such symptoms, none | Negative 100% | 2/120 |
| Q22 | String | poor, tired, normal, really low, block, low, lowest possible level, average, pretty average | Negative 98.10% | 4/67 |
| Q23 | String | same, normal, good, no, increased, unchanged | Positive 66.70% | 2/53 |
| Q24 | String | same, normal, no, decreased, unchanged | Negative 85.40% | 2/71 |
| Q25 | String | no, bone pain, headaches, body pain, swollen fingers, blurred eyes, body is killing, stomach ache, sore arm, sore neck | Positive 70.60% | 2/82 |
| Q26 | String | no, no such feelings, carrying my body, yes, feeling to be weighted down, just ok | Neutral 52.80% | 2/79 |
| Q27 | String | as usual, good, normal, the same level, pretty off | Negative 75.90% | 5/79 |

By analysing the rating and the satisfaction rate, we can continue to improve DEPRA to make it more interactive and encouraging for potential participants.

Appendix E in S1 Appendix provides the user satisfaction form rated by the participants.

## Measures and results

### Participants' profile

Table 4 shows the demography of the sample (N = 50) and the breakdown of the individual groups. The average age of the participants is 37.4 years, and more than half are male, 54% (27/50). Most participants are Middle Eastern 62% (31/50) followed by Asian 28% (14/50). Caucasians, Indians, and others represent 10% (5/50) of the participants.

In the first group (VU staff and HDR students), 50% (8/16) of the participants are females, 32% (5/16) are males, and 18% (3/16) prefer not to disclose their gender. The average age of the first group is 22.2 years. The majority of this group is Asian, 63% (10/16). The remainder of the first group of participants are Middle Eastern 25% (4/16) and others 12% (2/16).

The second group (Facebook friends) consists of 42% (12/28) females, 50% (14/28) males, and 8% (2/28) who prefer not to disclose their gender. The average age of this group is 38.6 years. The majority of this group is Middle Eastern 71% (20/28) with 22% (6/28) Asian, and others 7% (2/28).

The third group has the highest rate of dropout, which is to be expected. People found the chatbot on the page on their own, and due to their curiosity, they decided to interact. The organic group consists of 66% (4/6) females, 17% (1/6) males, and 17% (1/6) who prefer not to disclose their gender. The average age of this group is 37.2 years. The majority is Asian, 66% (4/6). The remainder of this group participants is Middle Eastern 17% (1/6) and others 17% (1/6).

### Participants depressive variables

In this study, we applied two methods for early depression detection scoring, namely, IDS-SR and QIDS-SR [43]. In the IDS-SR method, all 17 categories are considered namely mood, reactivity of mood, outlook (future), guilt, insomnia, suicidal ideation, anxiety, involvement, concentration, psychomotor, somatic, symptoms, general, pleasure, weight, interpersonal

**Table 4. Demographic variables of participants.**

| Demographic Variables | Group 1 (VU Staff and HDR Students) | Group 2 (Facebook Friends) | Group 3 (Organic) | Entire Sample (N = 50) |
|---|---|---|---|---|
| **Gender,n(%)** | | | | |
| Female | 8 (50%) | 12 (42%) | 4 (66%) | 22 (44%) |
| Male | 5 (32%) | 14 (50%) | 1 (17%) | 27 (54%) |
| Not Disclosed | 3 (18%) | 2 (8%) | 1 (17%) | 1 (2%) |
| **Ethnicity,n(%)** | | | | |
| Asian | 10 (63%) | 6 (22%) | 4 (66%) | 14 (28%) |
| Middle Eastern | 4 (25%) | 20 (71%) | 1 (17%) | 31 (62%) |
| Other | 2 (12%) | 2 (7%) | 1 (17%) | 5 (10%) |
| **Age Group,n(%)** | | | | |
| 20–29 | 3 (19%) | 5 (18%) | 1 (17%) | 18 (36%) |
| 30–39 | 3 (19%) | 6 (22%) | 3 (49%) | 12 (24%) |
| 40–49 | 8 (50%) | 16 (57%) | 1 (17%) | 15 (30%) |
| 50+ | 2 (12%) | 1 (3%) | 1 (17%) | 5 (10%) |

sensitivity, and genital symptoms. However, in QIDS-SR, the categories are divided into 9 groups of emotions. They are cited as sleep (sleep onset insomnia, mild-nocturnal insomnia, early morning insomnia and hypersomnia), mood, weight (appetite decreased, appetite increased, weight decrease, weight increase), concentration, guilt, suicidal ideation, interest, fatigue, and psychomotor changes (psychomotor slowing, psychomotor agitation). Eqs 1 and 2 are used to calculate the IDS-SR and QIDS-SR methods, respectively.

$$IDS_{TotalScore} = \sum_{i=1}^{27} MQ_i \qquad (1)$$

where $MQ_i$ is the manual score of the $i^{th}$ question.

$$QIDS_{TotalScore} = MQ22_{SomaticEnergy} + MQ1_{Mood} + MQ5_{Guilt} + MQ6_{Suicidal}$$

$$MQ11_{Interest} + MQ13_{Concentration} + MAX(MQ7_{InitialInsomnia}, MQ8_{MildInsomnia},$$

$$MQ9_{MorningInsomnia}, MQ10_{Hypersomnia}) + MAX(MQ21_{Appetite}, MQ26_{Weight}) \qquad (2)$$

$$+MAX(MQ14_{PsychomotorSlowing}, MQ15_{Agitation})$$

To summarize, for the IDS-SR total score, we added all the manual scores of each psychiatric question. All 27 manual values were added and the summation displays the overall value for the IDS-SR method.

Regarding QIDS-SR, there are four categories. First, the maximum value under three groups is calculated. The maximum value of insomnia is between initial insomnia, mild insomnia, morning insomnia and hypersomnia. The next step is the maximum value between appetite and weight. The last group of symptoms includes psychomotor slowing and agitation. After the calculation of the maximum values in each set, we simply add the manual scoring of semantic energy, guilt, suicidal ideation, interest and concentration. The summation of these groups gives the QIDS-SR total score.

The categories related to IDS-SR and QIDS-SR and the corresponding questions are summarized in Table 5a and 5b, respectively.

## Score analysis

According to the Inventory of Depressive Symptomatology (IDS) and the Quick Inventory of Depressive Symptomatology (QIDS) [45], the medical range for IDS-SR and QIDS-SR methods are shown in Table 6.

Regarding the IDS-SR scoring system, it is observed that around 30% (15 out of 50) of the participants were experiencing no depression and were healthy. As a result, no further follow-up session with the medical profession or a psychiatrist was recommended. Approximately the same number of participants had very severe symptoms 20% (10 out of 50) and moderate symptoms 22% (11 out of 50). Finally, the number of participants with severe symptoms 14% (7 out of 50) was the same as the number of participants with mild symptoms. Fig 7a displays the findings of the severity of depression within five groups—such as mild, moderate, no depression, severe and very severe—and the percentages which belong to each group of IDS-SR scoring system.

Regarding the QIDS-SR scoring system, the graph shows that around 32% (16 out of 50) of the participants were completely healthy. So, no further action was required to be taken by the participants in this group. 22% (11 out of 50) of the patients had very severe symptoms and required immediate medical advice and were asked to visit a medical professional as soon as

**Table 5. Symptoms categories and relevant questions of scoring systems.**

| (a). IDS-SR | |
|---|---|
| IDS-SR Symptoms | |
| **Category** | **Questions** |
| Mood | 1,3,4 |
| Outlook | 2 |
| Guilt | 5 |
| Suicidal | 6 |
| Insomnia | 7,8,9,10 |
| Interest | 11,12 |
| Concentration | 13 |
| Psychomotor Slowing | 14 |
| Agitation/Anxiety | 15,16,17 |
| Panic | 18 |
| Arousal | 19 |
| Gastro | 20 |
| Appetite | 21 |
| Somatic | 22,23,24 |
| Genital | 25 |
| Weight | 26 |
| Interpersonal Sensitivity | 27 |
| (b). QIDS-SR | |
| QIDS-SR Symptoms | |
| **Category** | **Questions** |
| Mood | 1 |
| Guilt | 5 |
| Suicidal | 6 |
| Interest | 11 |
| Concentration | 13 |
| Somatic Energy | 22 |
| Insomnia | 7,8,9,10 |
| Appetite/Weight | 21,26 |
| Slowing/Agitation | 14,15 |

possible to follow up on their health condition. 18% (9 out of 50) of the participants had severe and mild symptoms. The patients with severe symptoms were advised to attend a medical consultation when possible so the condition did not escalate but the participants with mild symptoms were not required to visit a health care professional, however they were asked to read

**Table 6. Comparison of total scores with regard to depression levels.**

| IDS-SR | QIDS-SR | Depression Level |
|---|---|---|
| Range between 0 to 13 | Range between 0 to 5 | No Depression, Healthy |
| Range between 14 to 25 | Range between 6 to 10 | Mild |
| Range between 26 to 38 | Range between 11 to 15 | Moderate |
| Range between 39 to 48 | Range between 16 to 20 | Severe |
| Range between 49 to 84 | Range between 21 to 27 | Very Severe |

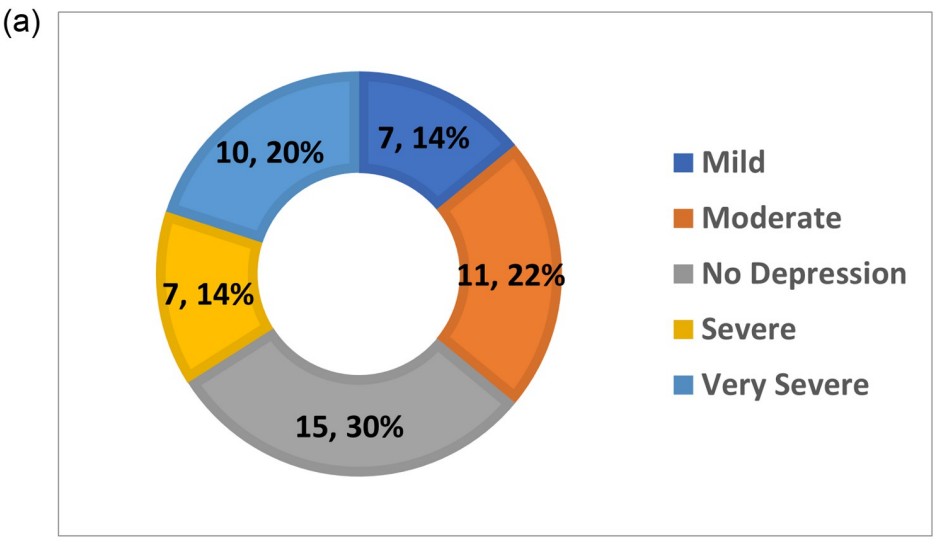

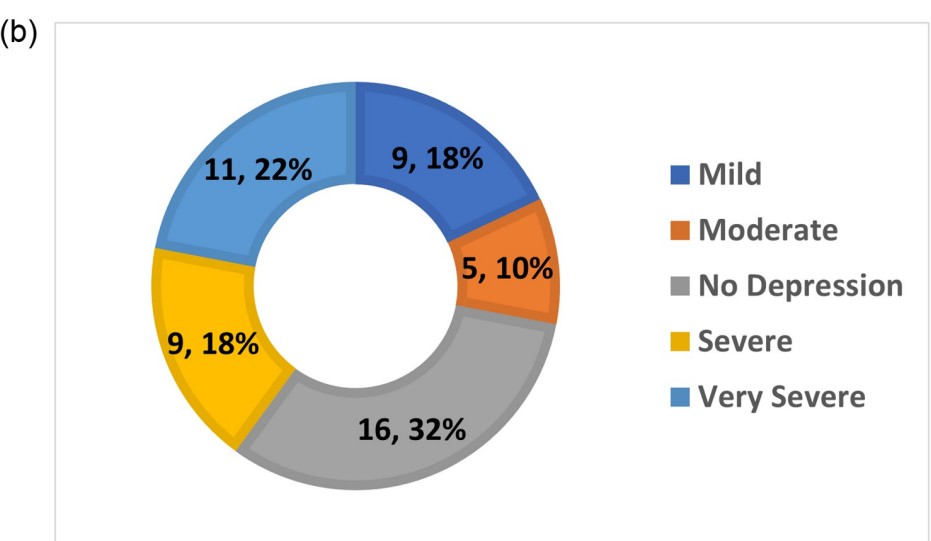

**Fig 7. Depression level statistics based on scoring systems (IDS-SR and QIDS-SR).** (**a**) IDS-SR. (**b**) QIDS-SR.

about their symptoms and be aware in case the symptoms increased. They were not considered to be at risk at this stage of the research. Lastly, 10% (5 out of 50) of the participants had moderate symptoms. Fig 7b displays the findings of the severity of depression within five groups—such as mild, moderate, no depression, severe and very severe—and the percentages which belong to each group of QIDS-SR scoring system.

Fig 8 compares the 50 participants scored by the IDS-SR against those scored by the QIDS-SR, showing that the overall trend of these two scoring systems is identical. However, there are minor differences to be noted. For example, 10 out of 50 participants were evaluated to have very severe depression by the IDS-SR scoring system while the QIDS-SR scoring system showed 11 of the 50 participants suffered from very severe depression. By applying both scoring systems and comparing them against each other, the overall analysis reflects a more accurate and self-explanatory outcome.

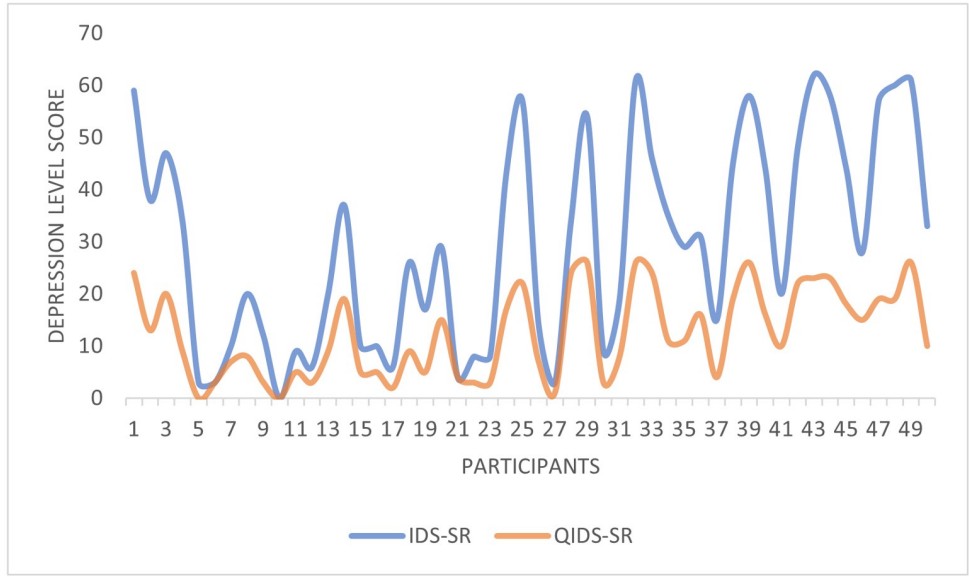

**Fig 8. Scoring system comparison against IDS-SR and QIDS-SR.**

Appendix F in S1 Appendix provides a link to the Microsoft Excel Worksheet with a comprehensive list of participants and their responses as well as the IDS-SR and QIDS-SR scores for the first 50 participants in the DEPRA chatbot interaction.

## User satisfaction and engagement

In this section, we detail the set of questions which were posed to the participants at the end of the data collection phase. A link was given to the participants as soon as they finished the DEPRA chatbot session. There are five linear scale questions where the participant must choose a response from a range of 1 (strongly disagree) to 5 (strongly agree) and one open-ended question in this rating assessment. Overall, we received 29 participants' rating form submissions from the survey.

The first question related to the ease of understanding and responding to the questions from the chatbot. 41.4% (12 out of 29) agreed that DEPRA's questions were easy to understand and respond to. 34.5% (10 out of 29) of the participants strongly agreed that DEPRA's questions were easy to understand. Only 24.1% (7 out of 29) of the participants felt neutral, neither agreeing nor disagreeing that DEPRA's questions were easy to understand. No participants indicated that they disagreed with the statement. Fig 9a illustrates the participants' responses to question 1.

The second question related to the time spent on this survey in comparison with a real psychiatrist session. The same number of participants 34.5% (10 out of 29) either agreed with or felt neutral in relation to the question about the time required to participate in the DEPRA conversation being less than the time required to see a psychiatrist in person. 27.6% (8 out of 29) strongly agreed that the time was managed better and only 3.4% (1 out of 20) strongly disagreed. Fig 9b illustrates the participants' responses to question 2.

The third question asks the participants to indicate their preference for the use of text messaging rather than talking to a psychiatrist. The same number of participants 31% (9 out of 29) either strongly agreed or felt neutral regarding their preference for text messaging rather than talking to a medical professional. The remainder of the participants 27.6% (8 out of 29) agreed

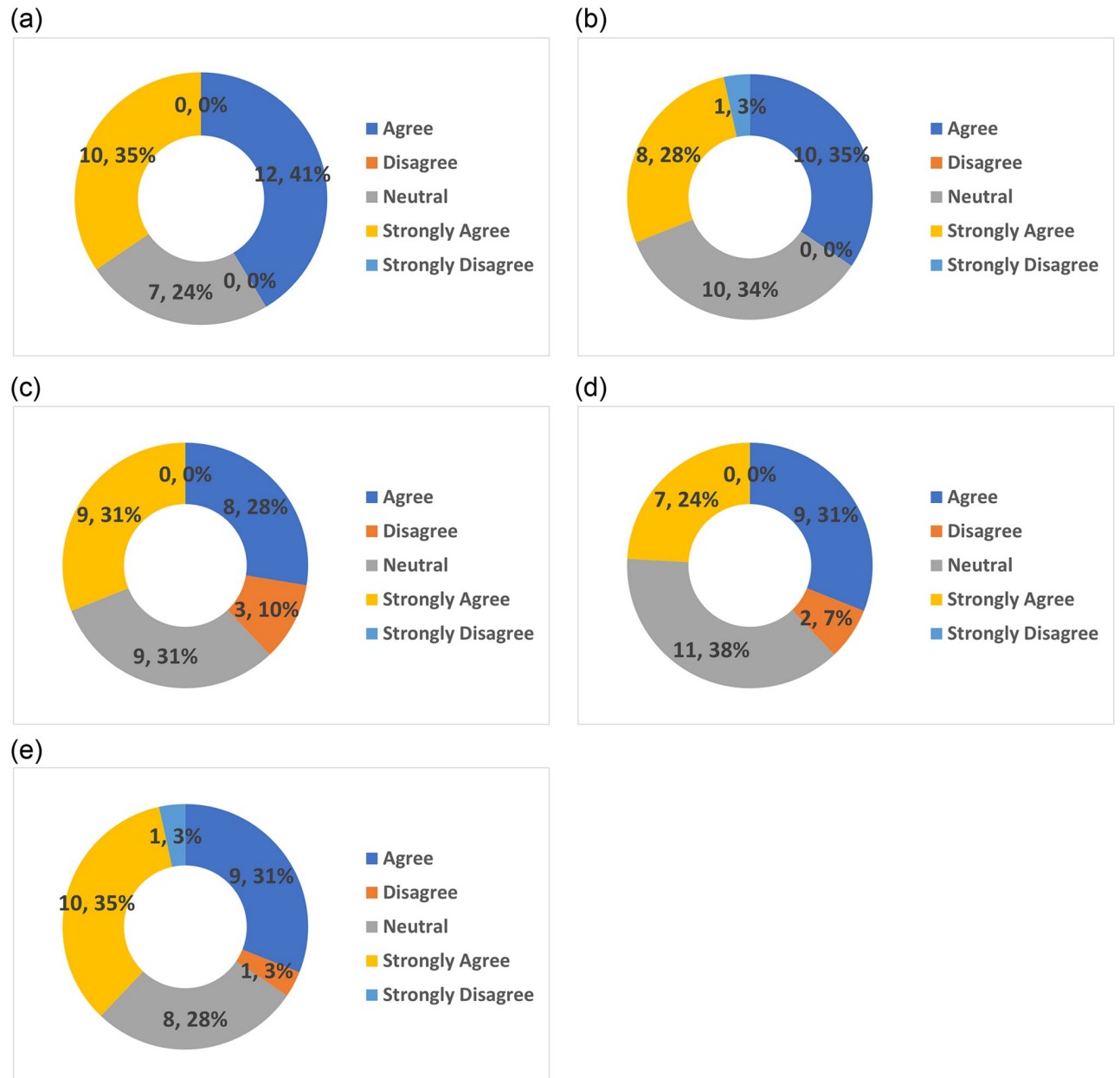

**Fig 9. Depression level statistics according to participants interactions.** (**a**) Participants' Response to Question 1. (**b**) Participants' Response to Question 2. (**c**) Participants' Response to Question 3. (**d**) Participants' Response to Question 4. (**e**) Participants' Response to Question 5.

that they preferred to send text messages rather than engage in a conversation with a psychiatrist. No participants strongly disagreed with the idea of text messaging compared to attending a real psychiatrist session, however, 10.3% (3 out of 29) disagreed. Fig 9c illustrates the participants' responses to question 3.

The fourth question asks the participants whether they felt that the sequence of questions directed the participant to reveal their level of depression. Most of the participants 37.9% (11 out of 29) were neutral, 31% (9 out of 29) agreed that sequence of the questions led the participant to reveal their level of depression and 24.1% (7 out of 29) strongly agreed. Only 6.9% (2

out of 29) disagreed with the idea that the sequence of questions led the participant to reveal their level of depression and no participants strongly disagreed. Fig 9d illustrates the participants' responses to question 4.

The last question asks the participants whether they are likely to recommend this survey to their friends or families. 34.5% (10 out of 29) strongly agreed that they would invite their friends and families to participate in the survey and 31% (9 out of 29) agreed. 27.6% (8 out of 29) of the participants were neutral and the same number of participants 3.4% (1 out of 29) disagreed and strongly disagreed. Fig 9e illustrates the participants' responses to question 5.

The open-ended question captured valuable comments of the participants such as "Following the pandemic, having access to a digital health system seems a necessity".

Table 7 summarizes the ratings of the 29 participants of the five questions and shows the average satisfaction rate of each participant's response.

In summary, the overall satisfaction rate was 3.95 out of 5 (79%) which suggests user satisfaction and engagement is at a promising rate. As shown in Table 7, most of the participants rated the questions, on average, 4 out of 5 (80%). Although this rate is promising, there is still room to improve the question design to increase the satisfaction rate of the participants and to

**Table 7. User satisfaction and engagement.**

| Participants | Q1 | Q2 | Q3 | Q4 | Q5 | Average Satisfaction Rate |
|---|---|---|---|---|---|---|
| Participant1 | 3 | 4 | 4 | 3 | 3 | 3.4 |
| Participant2 | 3 | 3 | 3 | 3 | 4 | 3.2 |
| Participant3 | 3 | 3 | 3 | 4 | 1 | 2.8 |
| Participant4 | 3 | 3 | 3 | 3 | 3 | 3.0 |
| Participant5 | 5 | 4 | 4 | 3 | 5 | 4.2 |
| Participant6 | 5 | 5 | 4 | 4 | 4 | 4.4 |
| Participant7 | 5 | 5 | 5 | 5 | 5 | 5.0 |
| Participant8 | 4 | 4 | 3 | 3 | 4 | 3.6 |
| Participant9 | 5 | 5 | 3 | 4 | 5 | 4.4 |
| Participant10 | 4 | 3 | 2 | 3 | 3 | 3.0 |
| Participant11 | 4 | 3 | 5 | 4 | 4 | 4.0 |
| Participant12 | 5 | 4 | 3 | 3 | 5 | 4.0 |
| Participant13 | 5 | 5 | 5 | 5 | 5 | 5.0 |
| Participant14 | 5 | 4 | 4 | 3 | 5 | 4.2 |
| Participant15 | 4 | 4 | 2 | 3 | 3 | 3.2 |
| Participant16 | 4 | 3 | 3 | 4 | 4 | 3.6 |
| Participant17 | 3 | 3 | 3 | 3 | 3 | 3.0 |
| Participant18 | 4 | 4 | 3 | 4 | 2 | 3.4 |
| Participant19 | 5 | 5 | 4 | 4 | 5 | 4.6 |
| Participant20 | 5 | 5 | 5 | 4 | 4 | 4.6 |
| Participant21 | 3 | 5 | 4 | 5 | 3 | 4.0 |
| Participant22 | 4 | 3 | 5 | 5 | 3 | 4.0 |
| Participant23 | 4 | 4 | 5 | 3 | 4 | 4.0 |
| Participant24 | 4 | 5 | 5 | 2 | 4 | 4.0 |
| Participant25 | 3 | 3 | 4 | 5 | 5 | 4.0 |
| Participant26 | 4 | 4 | 2 | 2 | 3 | 3.0 |
| Participant27 | 4 | 4 | 4 | 4 | 4 | 4.0 |
| Participant28 | 5 | 1 | 5 | 5 | 5 | 4.2 |
| Participant29 | 4 | 3 | 5 | 5 | 5 | 4.4 |

ensure they feel more connected to the questions, and have a higher rate of engagement with any set of questions.

## Discussions and future works

The objective of this study was to examine the feasibility and efficacy of using a conversational AI agent, which had been verified by psychiatrists, to engage in early depression detection. The results are based on self-reported symptoms from various perspectives. The methodology and results were consistent with the psychological interview guidelines. Our hypothesis was that the participants will be able to identify their depression symptoms at an early stage upon completion of the interactivity with DEPRA. The conversational flow was well designed under professional supervision. The outcomes from the user experience survey indicate the significance of the approach.

### Participant recruitment

To comply with the Victoria University ethics guidelines, we were only permitted to invite participants who were living in Australia and aged between 18 to 80 years old in this non-clinical trial. However, we believe this work will have a greater impact and will receive more widespread support if this article is published. Moreover, the discussion with the university has started and is scheduled for further development. The future target audience will vary in terms of their country of origin, ethnicity, age, and gender and will include those who potentially have mental health concerns.

### Participant dropout rates

This study comprises three participation groups, namely Group 1: staff and HDR students, Group 2: friends who interact via Facebook, and Group 3: the open access organic group. The total dropout rate as shown in Table 8 for each group from first contact was 79%, 48%, and 94%, respectively. The third group has the highest rate of dropout, which is to be expected. People who were curious found the DEPRA chatbot on the Facebook page on their own and decided to interact. However, many (91%) were unwilling to continue after reading the consent form. Moreover, we can make reasonable assumptions after a few informal interviews that the participants were worried about revealing their personal information when they required to sign the consent form, especially when the project did not have a solid manifestation, such as scientific articles that have already been published. The consent form dropout rate for Group 2 was the lowest at 15% due to the fact that the authors had the opportunity to explain the project in detail.

Another interesting finding is the dropout rate before the consent form was sent, where Group 3 had lowest dropout rate at 13% and Group 1 had highest at 43%. 70 out of the 85 invited participants reached the point of reading and signing the consent form. These results

**Table 8. Participants dropout rates.**

| | Group 1 (Staff & HDR) | Group 2 (Friends & Relatives) | Group 3 (Organic) |
|---|---|---|---|
| **Initial Participants Reached** | 75 | 54 | 95 |
| Dropout Rate (First Response) | 15 (20%) | 9 (17%) | 15 (16%) |
| Dropout Rate (Consent Form Sent) | 34 (43%) | 12 (27%) | 10 (13%) |
| Dropout Rate (Consent Form Receive) | 10 (38%) | 5 (15%) | 64 (91%) |
| **Total Dropout Rate** | 59 (79%) | 26 (48%) | 89 (94%) |

demonstrate that the depression detection chatbot has significant potential on the social media platform. A study by Google [45] also reported that the search interest for "therapists near me" hit record highs in 2021, and the phrase "why do I feel anxious for no reason" also hit an all-time high this year, spiking at more than 400%.

In summary, the characteristics of DEPRA make it an attractive platform on social media to help users who are concerned about their depression level. Furthermore, we believe this publication will increase the confidence level of volunteers to contribute to future studies.

## Participant satisfaction scales

We received 29 valid survey submissions which included responses to questions regarding the ease of answering questions in the questionnaire, the acceptance of the human-computer interaction, the interaction time, the question sequence in terms of depression severity level guidance, stress detection, and the liklihood that they will recommend the use of DEPRA to friends and family. It is noteworthy that 94% of all questionnaire responses were satisfactory in relation to DEPRA. This indicates that the majority of users believe that DEPRA is useful in relation to their depression detection and they are satisfied with the interaction. Although the number of participants was limited by geographic location and interpersonal relationships, a certain percentage of the total number of users was anonymous. Therefore, this satisfaction level is still informative in this preliminary work. In future work, we plan to involve participants from a wider range of ethnicities in the experiment to obtain more comprehensive results.

After we analyzed the results of each questionnaire, we made a very explorable observation. In Q3, we measured the users' perceptions of text messages and noted that three users returned negative responses. This question received the largest negative score of all the questions in the questionnaire. From this result, we speculate that text input becomes a relatively unreasonable request when the information described by the user involves a long paragraph. This is understandable as inputing lengthy text leads to problems such as mis-touching the send button, the input box not being able to display all the content of a lengthy text, the long time required to type a lengthy text causing the page response time to expire, etc. Therefore, we will use this user feedback as a reference to explore and study natural language processing in voice inputs and its processing in our future works.

Furthermore, we received 1/5 for the Q2 which explored whether the interaction with DEPRA was satisfactory to the users in terms of time spent compared with attending the physical psychiatrist session. We believe few factors may potentially impact the results including the quantity of the questions, the friendliness of contents for non-English speakers, the length of the questions, and the questions may require too much input from the users. Therefore, we genuinely believe that this questionnaire will provide a very high reference and benchmark value for DEPRA in its future studies.

## DEPRA chatbot engagement

We note that it is possible to make the chatbot more interactive and more responsive to the participants. The monitoring process of the research team proved that the design of the DEPRA chatbot can be improved to encourage more participants to answer the set of questions. For example, if the participant receives a short text message after answering each question, this will make the chatbot interaction more like a real psychiatrist session. This text message can encourage the participant to continue the process or it simply provides a sympathetic reaction to an optimistic or pessimistic response from the participant.

## Planning of validations

As a non-clinical trial study, this study also aims to identify whether DEPRA has the greatest probability of success, assess its safety, and build solid scientific foundations before transitioning to a future clinical development phase. Thus, this study created a realistic experimental environment and recruited local users to better validate the usability of DEPRA. The results of the experiment demonstrate that users were relatively more receptive to DEPRA and accepted the effectiveness of DEPRA for the early detection of mental health concerns. For the scientific validation of the questionnaire, we approached several psychologists and a team of experts in related professions to undertake an authoritative validation. In addition, the questionnaire will be improved according to professional guidance and will be reflected in the forthcoming works. Moreover, we are currently using the IDS-C and QIDS-C to validate the results. The work of the comparison group will continue to be developed, including and not limited to the automated scoring system, the comparison of the scoring system with the expert team's scoring results, and the comparison of the scoring results with the user's perception.

## Auto scoring

The current implementation of the DEPRA chatbot relies on the manual calculation of the depression score by checking the response manually and scoring each response separately. However, in the future, we will use machine learning and sentiment analyses algorithms to automatically check the responses and automatically calculate the depression score and return it to the participants immediately.

## Conclusions

Countries around the world are currently facing a pandemic so the need to access digital health care systems is more critical than at any time before. There is an urgent need for digital assessors to work with medical staff and assist with the collection of data to identify the population which needs professional consultation. In this research, we designed and implemented a DEPRA chatbot. A contemporary platform, Dialogflow, was used throughout the research. Dialogflow has practical tools that can be used to build conversational applications like chatbots which can be integrated in various environments such as Facebook Messenger. The Structured Interview Guide for the Hamilton Depression rating scale (SIGH-D) and the Inventory of Depressive Symptomatology (IDS) was followed to generate a set of questions required for data collection. SIGH-D and IDS contain instructions and topics for several mood symptoms confirmed by health care professionals. In this research, 50 participants completed the chatbot sessions and their responses were collected and analysed. To identify the participants' depression levels, two scoring systems were utilized, IDS-SR and QIDS-SR. The results for the IDS-SR scoring method showed that 30% of the participants were healthy, 14% had mild depression, 22% had moderate depression, 14% had severe depression, and 20% had very severe depression. The results for the QIDS-SR scoring method showed that 32% of the participants were healthy, 18% had mild depression, 10% had moderate depression, 18% had severe depression, and 22% had very severe depression. Finally, a user satisfaction form was completed by 29 out of 50 participants. The overall satisfaction rate was 3.95 out of 5 (79%), revealing a high rate of user satisfaction and engagement.

## Supporting information

**S1 Appendix.**
(PDF)

**S1 File.**
(DOCX)

## Author Contributions

**Conceptualization:** Payam Kaywan, Khandakar Ahmed, Ayman Ibaida.

**Formal analysis:** Payam Kaywan.

**Investigation:** Payam Kaywan.

**Methodology:** Payam Kaywan, Khandakar Ahmed, Ayman Ibaida.

**Supervision:** Khandakar Ahmed, Ayman Ibaida, Yuan Miao.

**Validation:** Payam Kaywan.

**Writing – original draft:** Payam Kaywan, Khandakar Ahmed.

**Writing – review & editing:** Khandakar Ahmed, Yuan Miao, Bruce Gu.

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
