## [Decision Letter · Decision Letter 0]

5 Apr 2022

PONE-D-22-04147Early Detection of Depression Using a Conversational AI Bot: a Non-Clinical TrialPLOS ONE

Dear Dr. Ahmed,

Thank you for submitting your manuscript to PLOS ONE. After careful consideration, we feel that it has certainly merit and this was acknowledged by both reviewers but it does not fully meet PLOS ONE’s publication criteria as it currently stands. Therefore, we invite you to submit a revised version of the manuscript that addresses the points raised during the review process. RThe major isue is that the organisation of the paper should be improved. 

We look forward to receiving your revised manuscript.

Kind regards,

Gilles van Luijtelaar, Ph.D.

Academic Editor

PLOS ONE

Journal Requirements:

2. We note that Figures 1, 2a, and 2b in your submission contain copyrighted images. All PLOS content is published under the Creative Commons Attribution License (CC BY 4.0), which means that the manuscript, images, and Supporting Information files will be freely available online, and any third party is permitted to access, download, copy, distribute, and use these materials in any way, even commercially, with proper attribution. For more information, see our copyright guidelines: http://journals.plos.org/plosone/s/licenses-and-copyright.

a. You may seek permission from the original copyright holder of Figures 1, 2a, and 2b to publish the content specifically under the CC BY 4.0 license. 

Additional Editor Comments:

It seems that there is lots of room for a better paper, including its orgnisation. On the other hand, both reviewers saw the merit of your work. Therefore I recommend "major revision".

Reviewers' comments:

Reviewer's Responses to Questions

**Comments to the Author**

1. Is the manuscript technically sound, and do the data support the conclusions?

Reviewer #1: Partly

Reviewer #2: Yes

2. Has the statistical analysis been performed appropriately and rigorously? 

Reviewer #1: No

Reviewer #2: Yes

3. Have the authors made all data underlying the findings in their manuscript fully available?

Reviewer #1: Yes

Reviewer #2: Yes

4. Is the manuscript presented in an intelligible fashion and written in standard English?

Reviewer #1: Yes

Reviewer #2: Yes

5. Review Comments to the Author

Reviewer #1: The concept of the paper, overall, is very compelling, and I agree that this method of detecting depression is important and relevant. Therefore, having this sort of work in the literature is important. However, there are several points about how the paper is currently structured that make it difficult to follow at times:

1). Overall, it is not clear whether this paper is meant as a proof-of-concept and feasibility statement or if it is meant to demonstrate that the chatbot is effective in detecting depression. For the latter to be properly demonstrated, several points must be addressed. First, it appears that the one of the rating scales used has been modified. While the modifications were approved by psychiatrists, they have not been scientifically validated. Second, there is no comparison group. To demonstrate that the chatbot is reliable, it should be compared to a well-validated measure of depression screening, ideally a clinician administered structured interview. Finally, there is no statistical analysis provided. If this is meant as a proof-of-concept, it would be interesting to flesh out some of the ideas presented, discuss next steps, and overall broaden the discussion about how this will move forward in the future.

2). The organization of the paper should be reviewed. There are times when information is presented in one section but really belongs in another. For example, the placement of the "objectives" section in the introduction might be better placed in the methods section. Also, it is best to refrain from editorializing in the results section. For example, there is a discussion about the drop-out rate in one of the groups. This point is very interesting, particularly if you believe that this is the group from which most of the real-life sample will be drawn. It deserves more attention and is better placed in the discussion.

3). There are several points made in the introduction that either need references or need clarification that it is the writer's opinion.

4). The novelty of the chatbot is claimed but not fully supported. The writer discussed other studies that have examined artificial intelligence use to detect psychiatric symptoms. It appears that this particular chatbot focuses more on depression or has an easier platform, but it is not entirely clear how or why.

5). The methods section would benefit from more clarification and definition of terms. Most importantly, many readers may need "chatbot" defined. The methods section overall assumes a level of familiarity with computer programming that many, but not all, readers will have. Terms such as "front-end," "back-end," "intent," and "dynamic response" should be defined. Further, the writer asserts that Facebook messenger is a trustworthy platform. This assumption is not entirely clear, and the security of the platform is not well-defined.

6). The language in the results section would benefit from clarification. There is a lot of talk about the HDRS scale earlier in the paper, and I do not see any results reported.

7). The discussion and the conclusions would benefit from being expanded. Currently, these sections are largely a repetition of the data collected. It would be interesting to hear what the writer's thoughts about the data and next steps are.

8). The writer notes that a limitation in the study is that it is not generalizable because it only included an Australian sample. I would add that even more importantly, the writer heavily recruited family and friends as participants. This method of recruitment introduces substantial bias around the drop-out rate and the satisfaction scales

9). The tables and charts need clarification and should have a caption, especially when not "in-line" for the text.

10). The writer states that the participants are "keen on revealing their moods" several times throughout the paper. While it seems to be a reasonable assumption, there are no data or objective measures presented to support this claim

11). The writer refers to the study as a "non-clinical" study on several occasions. This term is confusing. I think what is meant here is that the sample is drawn from a non-clinical group. This point should be clarified as well.

Overall, the concept is very compelling, and I look forward to learning more about this area of study.

Reviewer #2: Overall, this is a strong piece of research with very interesting and timely findings! A few thoughts to strengthen this manuscript:

1. A general proofread for grammar and readability issues is needed. For example, on page 2 in the Abstract Section, the sentence "First group comprises professional academic staff and HDR candidates, the second and third groups comprise relatives, friends, and volunteers who were recruited via social media promotions" could be more clearly stated. Further, on the same page another sentence begins: "computer-assisted mental health (CAMH)" without proper capitalization.

2. Under the Related Works section and Psychological Intervention subsection, mention briefly any notable results of the studies you cite, as you do in the following Early Depression Detection Subsection. For example, were these projects effective?

3. The Participant Recruitment on page 8 needs clarification. For example, after reading "We used three channels for recruitment (Fig 6) - the first group was university 208 students and university professional staff either full-time, part-time or causal" and looking at Figure 6, it is still unclear how participants were recruited into this study at various points.

4. Further, in your abstract you mention that participants "included academics and HDR candidates who are conscious, vigilant and have a clear understanding of the questions." Describe how this was assessed in the Participant Recruitment section.

5. Under the Limitations section on page 14, it would be helpful if you could speak to any potential bias relating to recruitment of the research team's friends and family as participants for this study, as compared to the general Australian public.

6. In your introduction you posit the following hypothesis: "We predict that the absence of a booking requirement and the flexibility of participating on their own schedule will eliminate an impediment to obtaining preparatory help and avoiding social embarrassment." Please note how that factors in with the overall conclusions of this study on page 15?

6. PLOS authors have the option to publish the peer review history of their article (what does this mean?). If published, this will include your full peer review and any attached files.

Reviewer #1: No

Reviewer #2: No

---

## [Author Response · Author response to Decision Letter 0]

20 Jun 2022

Responses to Reviewer #1’s Comments

Comment 1). Overall, it is not clear whether this paper is meant as a proof-of-concept and feasibility statement or if it is meant to demonstrate that the chatbot is effective in detecting depression. For the latter to be properly demonstrated, several points must be addressed. First, it appears that the one of the rating scales used has been modified. While the modifications were approved by psychiatrists, they have not been scientifically validated. Second, there is no comparison group. To demonstrate that the chatbot is reliable, it should be compared to a well-validated measure of depression screening, ideally a clinician administered structured interview. Finally, there is no statistical analysis provided. If this is meant as a proof-of-concept, it would be interesting to flesh out some of the ideas presented, discuss next steps, and overall broaden the discussion about how this will move forward in the future.

Response 1).

We appreciate the reviewer for pointing this out and providing constructive guidance. The paper is meant as a proof-of-concept and aims to provide a feasible solution with technological functionalities by demonstrating how chatbot works in depression detection. 

Furthermore, this study is a simplicity preliminary work with the convenience of use for our focus group audience since most of them are the first time to try on this approach. We have amended the paper with more comprehensive further plan. The next step following up with this preliminary work will include a scientific validation where the response will be sent to psychologist and compare with the result. 

Summary of changes in the context are as follows:

Page 1, Abstract, Objectives

This study aims to understand the feasibility and efficacy of using AI-enabled chatbots in the early detection of depression. This is a preliminary work which provides a simple and convenient solution for the focus group. Furthermore, the insights gained from the results are discussed and a comprehensive future plan is proposed with scientific validations.

Line 47 - 73, Page 2, Introduction

In summary, the community is currently open and positive on artificial intelligence and chatbots to help with mental health concerns. However, early detection before the onset of the disease is critical but is often overlooked. With this motivation, we proposed and developed a non-clinical chatbot, DEPRA, with social attributes to help users with mental health concerns and detect the illness in its early stage. This is a preliminary study which aims to provide a simple rather than a complex platform to validate its feasibility and efficacy.

DEPRA is capable of assisting participants examine their mental health conditions as a reference so that medical professionals can assist patients who are suffering from depression

• It was implemented following a conversation flow based structured early detection depression interview guide for the Hamilton Depression Scale (SIGH-D) [21] and Inventory of Depressive Symptomatology (IDS-C) [22] drawing on the experience of a group of psychiatrists. This makes the DEPRA chatbot a unique chatbot in that it shares the same set of clinically proved questions in the design of its questions for interaction with participants.}

• The validity of the questions used in the DEPRA chatbot was measured by applying closed group participation. This was arranged by designing open-ended questions for the participants. DEPRA is integrated with Facebook Messenger so, participants can interact with the chatbot through social media to share their responses.

• The questions asked by the DEPRA chatbot take approximately 30 minutes to complete. As the results suggest, the participants' responses indicate high satisfaction rates. They reported that the questions were easy to comprehend and respond to, it was not as time consuming as attending a face-to-face psychiatrist session, and they preferred the option to send text messages via social media platforms compared to talking to a psychiatrist in a consultation session.

Line 643 – 658, Page 23, Discussion and Future Works, Planning of Validation

As a non-clinical trial study, this study also aims to identify whether DEPRA has the greatest probability of success, assess its safety, and build solid scientific foundations before transitioning to a future clinical development phase. Thus, this study created a realistic experimental environment and recruited local users to better validate the usability of DEPRA. The results of the experiment demonstrate that users were relatively more receptive to DEPRA and accepted the effectiveness of DEPRA for the early detection of mental health concerns. For the scientific validation of the questionnaire, we approached several psychologists and a team of experts in related professions to undertake an authoritative validation. In addition, the questionnaire will be improved according to professional guidance and will be reflected in the forthcoming works. Moreover, we are currently using the IDS-C and QIDS-C to validate the results. The work of the comparison group will continue to be developed, including and not limited to the automated scoring system, the comparison of the scoring system with the expert team's scoring results, and the comparison of the scoring results with the user's perception.

Line 567 – 578, Page 22, Discussion and Future Works, Participant Recruitment

To comply with the Victoria University ethics guidelines, we were only permitted to invite participants who were living in Australia and aged between 18 to 80 years old in this non-clinical trial. However, we believe this work will have a greater impact and will receive more widespread support if this article is published. Moreover, the discussion with the university has started and is scheduled for further development. The future target audience will vary in terms of their country of origin, ethnicity, age, and gender and will include those who potentially have mental health concerns.

Line 576 – 599, Page 22, Discussion and Future Works, Participant Dropout Rates

This study comprises three participation groups, namely Group 1: staff and HDR students, Group 2: friends who interact via Facebook, and Group 3: the open access organic group. The total dropout rate for each group from first contact was 79%, 48%, and 94%, respectively. The third group has the highest rate of dropout, which is to be expected. People who were curious found the DEPRA chatbot on the Facebook page on their own and decided to interact. However, many (91%) were unwilling to continue after reading the consent form. Moreover, we can make reasonable assumptions after a few informal interviews that the participants were worried about revealing their personal information when they required to sign the consent form, especially when the project did not have a solid manifestation, such as scientific articles that have already been published. The consent form dropout rate for Group 2 was the highest at 15% due to the fact that the authors had the opportunity to explain the project in detail.

Another interesting finding is the dropout rate before the consent form was sent, where Group 3 had lowest dropout rate at 18% and Group 1 had highest at 65%. 70 of the 85 invited participants reached the point of reading and signing the consent form. These results demonstrate that the depression detection chatbot has significant potential on the social media platform. A study by Google [45] also reported that the search interest for “therapists near me” hit record highs in 2021, and the phrase "why do I feel anxious for no reason" also hit an all-time high this year, spiking at more than 400%

In summary, the characteristics of DEPRA make it an attractive platform on social media to help users who are concerned about their depression level. Furthermore, we believe this publication will increase the confidence level of volunteers to contribute to future studies.

Line 601 – 632, Page 23, Discussion and Future Works, Participant Satisfaction Scales

We received 29 valid survey submissions which included responses to questions regarding the ease of answering questions in the questionnaire, the acceptance of the human-computer interaction, the interaction time, the question sequence in terms of depression severity level guidance, stress detection, and the liklihood that they will recommend the use of DEPRA to friends and family. It is noteworthy that 94\\% of all questionnaire responses were satisfactory in relation to DEPRA. This indicates that the majority of users believe that DEPRA is useful in relation to their depression detection and they are satisfied with the interaction. Although the number of participants was limited by geographic location and interpersonal relationships, a certain percentage of the total number of users was anonymous. Therefore, this satisfaction level is still informative in this preliminary work. In future work, we plan to involve participants from a wider range of ethnicities in the experiment to obtain more comprehensive results.}

After we analyzed the results of each questionnaire, we made a very explorable observation. In Q3, we measured the users’ perceptions of text messages and noted that three users returned negative responses. This question received the largest negative score of all the questions in the questionnaire. From this result, we speculate that text input becomes a relatively unreasonable request when the information described by the user involves a long paragraph. This is understandable as inputing lengthy text leads to problems such as mis-touching the send button, the input box not being able to display all the content of a lengthy text, the long time required to type a lengthy text causing the page response time to expire, etc. Therefore, we will use this user feedback as a reference to explore and study natural language processing in voice inputs and its processing in our future works.

Furthermore, Q2 explored whether the interaction with DEPRA was satisfactory to users in terms of the time spent compared with attending the psychiatrist session. This is the only question for which we received a score of 1/5. Possible reasons for this result may be the number of questions, whether the content is easy to understand for non-first language English speakers, whether the questions are too long, or whether the user needs to provide more feedback for some questions, etc. This questionnaire will also provide a very high reference value for DEPRA in its future work.

Comment 2). The organization of the paper should be reviewed. There are times when information is presented in one section but really belongs in another. For example, the placement of the "objectives" section in the introduction might be better placed in the methods section. Also, it is best to refrain from editorializing in the results section. For example, there is a discussion about the drop-out rate in one of the groups. This point is very interesting, particularly if you believe that this is the group from which most of the real-life sample will be drawn. It deserves more attention and is better placed in the discussion.

Response 2).. 

We appreciate the reviewer very much for pointing this out. In the revised version, we remove “Objective” section from introduction, move to “Methodologies” section as an introductory paragraph. Furthermore, we also add some more explanation into the paragraph for better reading experience. 

Summary of changes in the context are as follows:

Line 229 – 252, Page 7, Methodologies

The objective of this study is to consider the efficacy and feasibility of applying an AI-based chatbot for early depression detection. DEPRA follows the textual conversations and interactions between the participants and the agent. Early depression detection is the core focus of this research. We are aware that the early detection of depression presents significant possibilities for treating patients and providing support to vulnerable members of society. DEPRA eliminates dependence on a limited set of multiple-choice replies in favour of open-ended responses, allowing participants to express their thoughts and sentiments spontaneously and without hesitation. DEPRA assesses mood, guilt, suicidal tendency, insomnia, agitation or retardation, anxiety, weight change, and bodily symptoms to diagnose depression levels.}

To determine the efficacy of utilising a chatbot for mass depression screening, we recruited 50 Australian residents. We hypothesise that individuals will feel at ease and have no qualms about interacting with an AI chatbot rather than an actual therapist. We integrated DEPRA into a Facebook page and enabled participants to contribute at their own pace and convenience. We predict that the absence of a booking requirement and the flexibility of participating on their own schedule will eliminate any impediment to obtaining preparatory help and avoiding social embarrassment. Furthermore, we assume that substituting open-ended replies for multiple choice enables individuals to avoid conscious reactions and helps us to capture a participant's genuine situation. To examine the postulations, we directed participants to complete a concise user experience survey after the main session using the same platform. At the completion of the participation, DEPRA scored the replies and classified the participants into five groups based on their SIGH-D and IDS-C scores: nil, mild, moderate, severe and very severe depression.

Line 643 – 658, Page 23, Discussion and Future Works, Planning of Validation

Same as above

Line 567 – 578, Page 22, Discussion and Future Works, Participant Recruitment

Same as above

Line 576 – 599, Page 22, Discussion and Future Works, Participant Dropout Rates

Same as above

Line 601 – 632, Page 23, Discussion and Future Works, Participant Satisfaction Scales

Same as above

Comment 3). There are several points made in the introduction that either need references or need clarification that it is the writer's opinion.

Response 3). 

We appreciate reviews for pointing this out. We have revisited the entire paper and added relevant references. 

Following reference has been added into this article:

[8] Carroll KM, Kiluk BD, Nich C, Gordon MA, Portnoy GA, Marino DR, et al. Computer-assisted delivery of cognitive-behavioral therapy: efficacy and durability of CBT4CBT among cocaine-dependent individuals maintained on methadone. American journal of Psychiatry. 2014;171(4):436–444.

[12]. Siedlikowski S, No ¨el LP, Moynihan SA, Robin M, et al. Chloe for COVID-19:Evolution of an Intelligent Conversational Agent to Address Infodemic Management Needs During the COVID-19 Pandemic. Journal of Medical Internet Research. 2021;23(9):e27283. 730

[13]. Jeyanthi PM. INDUSTRY 4. O: The combination of the Internet of Things (IoT) and the Internet of People (IoP). Journal of Contemporary Research in Management. 2018;13(4). 733

[14]. Park DM, Jeong SS, Seo YS. Systematic Review on Chatbot Techniques and Applications. Journal of Information Processing Systems. 2022;18(1):26–47. 735

[15]. Tiwari S, Bansal A. Domain-Agnostic Context-Aware Framework for Natural Language Interface in a Task-Based Environment. In: 2021 IEEE 45th Annual Computers, Software, and Applications Conference (COMPSAC). IEEE; 2021. p.15–20.

[16]. Melo G, Alencar P, Cowan D. A cognitive and machine learning-based software development paradigm supported by context. In: 2021 IEEE/ACM 43rd International Conference on Software Engineering: New Ideas and Emerging Results (ICSE-NIER). IEEE; 2021. p. 11–15. 743

[17]. Hassett A, Green C, Zundel T. Parental involvement: a grounded theory of the role of parents in adolescent help seeking for mental health problems. Sage open. 2018;8(4):2158244018807786.

[20]. Fulmer R, Joerin A, Gentile B, Lakerink L, Rauws M. Using psychological artificial intelligence (Tess) to relieve symptoms of depression and anxiety: randomized controlled trial. JMIR mental health. 2018;5(4):e64. doi:10.2196/mental.978.

[23]. P ´erez JQ, Daradoumis T, Puig JMM. Rediscovering the use of chatbots in education: A systematic literature review. Computer Applications in Engineering Education. 2020;28(6):1549–1565.

[25]. Ly KH, Ly AM, Andersson G. A fully automated conversational agent for promoting mental well-being: A pilot RCT using mixed methods. Internet Interventions. 2017;10:39–46. doi:https://doi.org/10.1016/j.invent.2017.10.002. 773

[26]. Sharma B, Puri H, Rawat D. Digital psychiatry-curbing depression using therapy chatbot and depression analysis. In: 2018 Second International Conference on Inventive Communication and Computational Technologies (ICICCT). IEEE; 2018. p. 627–631

[34]. Kamita T, Ito T, Matsumoto A, Munakata T, Inoue T. A chatbot system for mental healthcare based on SAT counseling method. Mobile Information Systems. 2019;2019. 802

[35]. Nutt A. The Woebot will see you now. the rise of chatbot therapy: Washington Post. 2017;. 

[36]. Fitzpatrick KK, Darcy A, Vierhile M. Delivering cognitive behavior therapy to young adults with symptoms of depression and anxiety using a fully automated conversational agent (Woebot): a randomized controlled trial. JMIR mental health. 2017;4(2): e7785.

[42]. Kim JY, Lee KM, Park SH. Evolution of revealing emotions. Physica A:Statistical Mechanics and its Applications. 2022;597:127268. Doi: https://doi.org/10.1016/j.physa.2022.127268.

[43]. Xu L, Zhou X, Gadiraju U. Revealing the role of user moods in struggling search tasks. In: Proceedings of the 42nd International ACM SIGIR Conference on Research and Development in Information Retrieval; 2019. p. 1249–1252.

[46]. Forman EM, Kerrigan SG, Butryn ML, Juarascio AS, Manasse SM, Onta ~n ´on S, et al. Can the artificial intelligence technique of reinforcement learning use continuously-monitored digital data to optimize treatment for weight loss? Journal of behavioral medicine. 2019;42(2):276–290. June 10, 2022 28/29

Comment 4). The novelty of the chatbot is claimed but not fully supported. The writer discussed other studies that have examined artificial intelligence use to detect psychiatric symptoms. It appears that this particular chatbot focuses more on depression or has an easier platform, but it is not entirely clear how or why.

Response 4). 

We appreciate the reviewer very much for pointing this out. We revised some existing sentences and added further explanation paragraphs in serval sections.

Summary of changes in the context are as follows:

Line 47 - 73, Page 3, Introduction

In summary, the community is currently open and positive on artificial intelligence and chatbots to help with mental health concerns. However, early detection before the onset of the disease is critical but is often overlooked. With this motivation, we proposed and developed a non-clinical chatbot, DEPRA, with social attributes to help users with mental health concerns and detect the illness in its early stage. This is a preliminary study which aims to provide a simple rather than a complex platform to validate its feasibility and efficacy.

DEPRA is capable of assisting participants examine their mental health conditions as a reference so that medical professionals can assist patients who are suffering from depression

• It was implemented following a conversation flow based structured early detection depression interview guide for the Hamilton Depression Scale (SIGH-D) [21] and Inventory of Depressive Symptomatology (IDS-C) [22] drawing on the experience of a group of psychiatrists. This makes the DEPRA chatbot a unique chatbot in that it shares the same set of clinically proved questions in the design of its questions for interaction with participants.}

• The validity of the questions used in the DEPRA chatbot was measured by applying closed group participation. This was arranged by designing open-ended questions for the participants. DEPRA is integrated with Facebook Messenger so, participants can interact with the chatbot through social media to share their responses.

• The questions asked by the DEPRA chatbot take approximately 30 minutes to complete. As the results suggest, the participants' responses indicate high satisfaction rates. They reported that the questions were easy to comprehend and respond to, it was not as time consuming as attending a face-to-face psychiatrist session, and they preferred the option to send text messages via social media platforms compared to talking to a psychiatrist in a consultation session.

Line 229 – 252, Page 7, Methodologies

The objective of this study is to consider the efficacy and feasibility of applying an AI-based chatbot for early depression detection. DEPRA follows the textual conversations and interactions between the participants and the agent. Early depression detection is the core focus of this research. We are aware that the early detection of depression presents significant possibilities for treating patients and providing support to vulnerable members of society. DEPRA eliminates dependence on a limited set of multiple-choice replies in favour of open-ended responses, allowing participants to express their thoughts and sentiments spontaneously and without hesitation. DEPRA assesses mood, guilt, suicidal tendency, insomnia, agitation or retardation, anxiety, weight change, and bodily symptoms to diagnose depression levels.}

To determine the efficacy of utilising a chatbot for mass depression screening, we recruited 50 Australian residents. We hypothesise that individuals will feel at ease and have no qualms about interacting with an AI chatbot rather than an actual therapist. We integrated DEPRA into a Facebook page and enabled participants to contribute at their own pace and convenience. We predict that the absence of a booking requirement and the flexibility of participating on their own schedule will eliminate any impediment to obtaining preparatory help and avoiding social embarrassment. Furthermore, we assume that substituting open-ended replies for multiple choice enables individuals to avoid conscious reactions and helps us to capture a participant's genuine situation. To examine the postulations, we directed participants to complete a concise user experience survey after the main session using the same platform. At the completion of the participation, DEPRA scored the replies and classified the participants into five groups based on their SIGH-D and IDS-C scores: nil, mild, moderate, severe and very severe depression.

Line 199 – 226, Page 6, Early Depression Detection, Related Works

Fulmer et al [20] proposed Tess which the interactions generated by about 354 participants with Tess depressions modules were considered to understand chatbot usage within the modules. In the research, the researchers conducted a randomized controlled trial to assess impact of using an integrative therapeutical AI agent, Tess, a mental health chatbot that provides self-help chats through text messages and evaluated its efficacy in reducing self-identified symptoms of depression and anxiety in college students. The study participants engaged in natural conversations with Tess via Facebook Messenger. No demographic variables of the participants were collected, so no data for this part of the experiment is available. Tess is able to react to the participants' emotional needs by analysing the conversational content either through textual interaction or by Facebook Messenger sampling. Tess is designed to play the role of a traditional therapist; however, it is not a replacement for face-to-face therapy by a professional. The experiment identified several limitations in the use of Tess. First, it is not possible to determine why a user stops interacting with Tess, for example, were they redirected to a different module, or did they simply lose interest in continuing with the interactions etc. Also, of the 354 participants, two claimed there were technical issues in the design of Tess and they stated that they could not finalize their interaction within the experiment and continue the sessions due yo these issues.

Our proposed DEPRA chatbot falls under this bot family category as a Depression Detection Bot, the aim of which is to circumvent some of the existing challenges that have been discussed in this section. The main difference between this research and the related works conducted so far is that the DEPRA chatbot gathers data based on a series of psychologically approved questions and its purpose is to triage and detect the early signs of depression. However, the available chatbots mostly focus on the depression disorder itself and try to find a better cure. In other words, the DEPRA chatbot aims to assist medical professionals detect and cure depression in its early stage and it does not aim to replace a medical professional. DEPRA was implemented to further investigate the application of AI to interact with users, collect their responses to questions on depression symptoms and identify depression and estimate the level of its severity.

Comment 5). The methods section would benefit from more clarification and definition of terms. Most importantly, many readers may need "chatbot" defined. The methods section overall assumes a level of familiarity with computer programming that many, but not all, readers will have. Terms such as "front-end," "back-end," "intent," and "dynamic response" should be defined. Further, the writer asserts that Facebook messenger is a trustworthy platform. This assumption is not entirely clear, and the security of the platform is not well-defined.

Response 5). 

Thanks for highlighting this point. We have added a new table (Table 1. Descriptions of Technical Terms) for better reading experience. Furthermore, we added new paragraph to further explain how and why Facebook Messenger could provide a trustworthy platform. 

Summary of changes in the context are as follows:

Line 28 – 38, Page 3, Introductions

A widely used CAMH application is the chatbot, also known as a conversational agent [12]. A chatbot is a computer application that replicates and processes human dialogue. It allows humans to communicate with a digital assistant as if they were conversing with an actual human [13]. A data-driven chatbot, also referred to as a virtual assistant is more interactive, sophisticated and customized. These chatbots are context-aware and use natural-language understanding (NLU) [14], natural language processing (NLP) [15], and machine learning [16] to learn the user inputs and generate output. In the longer term, a chatbot can employ analytics and predictive intelligence to provide personalization based on user profiles and previous activity. Over time, digital assistants may learn a user's preferences, provide recommendations, and even foresee requirements. They can start dialogues in addition to monitoring data and intent.

Table 1. Descriptions of Technical Terms, Page 10, Methodologies

Line 270 – 279, Page 8, Architecture of DEPRA, Methodologies

Facebook Messenger provides a convenient environment for participation, it is easy to access the DEPRA page, and the interaction window is designed internally by Facebook. Facebook Messenger also provides a trustworthy platform for users where various data security and privacy techniques are used, such as AI to identify unusual behavioural patterns correlated with harmful activities, reduce the spread of viral misinformation and harmful content, detecting impersonators, and enhancing safeguards for users under 18. Furthermore, the platform also provides end-to-end encryptions which reinforce safety and security and have increasingly become the standard expectation of users for their preferred communications platforms.

Comment 6). The language in the results section would benefit from clarification. There is a lot of talk about the HDRS scale earlier in the paper, and I do not see any results reported.

Response 6). 

We appreciate the reviewer very much for pointing this out. HDRS is a board term of the rating scale under Hamilton Interview Guide (HIG). the Structured Interview Guide for the HDRS (SIGH-D) was developed to standardize the manner of administration of the scale. This study focuses to abstract questioners from SIGH-D. Therefore, we have revised the paragraph to avoid further confusions and improve the readability. 

Summary of changes in the context are as follows:

Line 302 – 316, Page 9, Conversation Desgin, Methodologies

DEPRA uses a structured HIG interview guide developed by Williams et al. [21] in designing DEPRA’s conversation. The base of this guide is one of the most widely used clinician-administered depression scales, namely the Structured Interview Guide for the Hamilton Depression Rating Scale (SIGH-D) [39]. The SIGH-D standardizes the manner of administration and scoring of the scale from the original scaling system, the Hamilton Depression Rating Scale (HDRS) [40] [41]. The original guide identified twenty-one items to measure, though Hamilton himself only recommended using the first seventeen since the last four items are loosely associated with the common symptoms of depression. Initially, SIGH-D was designed for hospital in-patients, but later, experts devised many versions. All seventeen items pertain to symptoms that a depressed person could have experience over the past few weeks. SIGH-D was originally unstructured with the least general information for rating individual items. This gap motivates many works focusing on producing structured or semi-structured interview guides. HIG has thirty-one questions to measure seventeen items with 48 sub-questions for further comprehension and each question has four options from which to choose.

Comment 7). The discussion and the conclusions would benefit from being expanded. Currently, these sections are largely a repetition of the data collected. It would be interesting to hear what the writer's thoughts about the data and next steps are.

Response 7). 

We appreciate the reviewer very much for the constructive comments and feedbacks. We have add more subsections and paragraphs in the Discussion and Future Works section. 

Summary of changes in the context are as follows:

Line 643 – 658, Page 23, Discussion and Future Works, Planning of Validation

Same as above

Line 576 – 599, Page 22, Discussion and Future Works, Participant Dropout Rates

Same as above

Line 601 – 632, Page 23, Discussion and Future Works, Participant Satisfaction Scales

Same as above

Comment 8). The writer notes that a limitation in the study is that it is not generalizable because it only included an Australian sample. I would add that even more importantly, the writer heavily recruited family and friends as participants. This method of recruitment introduces substantial bias around the drop-out rate and the satisfaction scales.

Response 8). 

We appreciated the reviewer for pointing this out. The region and participants recruitment has been discussed further in the revision paper. 

Summary of changes in the context are as follows:

Line 643 – 658, Page 23, Discussion and Future Works, Planning of Validation

Same as above

Line 567 – 578, Page 22, Discussion and Future Works, Participant Recruitment

Same as above

Line 576 – 599, Page 22, Discussion and Future Works, Participant Dropout Rates

Same as above

Line 601 – 632, Page 23, Discussion and Future Works, Participant Satisfaction Scales

Same as above

Comment 9). The tables and charts need clarification and should have a caption, especially when not "in-line" for the text.

Response 9). 

We appreciate the reviewer to point this out. We have revisited every figure and table and added more clarifications for the captures as well as in-line descriptions. 

Summary of changes in the context are as follows:

Figure 1, Schematic Diagram of DEPRA - Design of the Implementation, Page 7, Architecture of DEPRA, Methodologies

Line 265, Page 8, Architecture of DEPRA, Methodologies

Fig 1 shows the schematic diagram of DEPRA. In this Figure the implementation and design of DEPRA chatbot is visualized.

Figure 3, Schema/E-R diagram of the central database, Page 10, Architecture of DEPRA, Methodologies

Line 297 - 300, Page 9, Architecture of DEPRA, Methodologies

We validate the response received from the user and store it in the central database (Fig 3 shows the schema/E-R diagram of the central database which includes all the tables in DB and the way they are interrelated) and we defined a number of functions in the fulfillment.

Figure 6, Participant Recruitment Flow – 3 Groups indicating Statistics of Complete Response and Drop out Rate, Page 15, Participant Recruitment, Methodologies

Figure 7, Depression Level Statistics based on Scoring Systems (IDS-SR and QIDS-SR), Page 19, Score Analysis, Measures and Results

Line 297 - 300, Page 9, Score Analysis, Measures and Results

We validate the response received from the user and store it in the central database (Fig 3 shows the schema/E-R diagram of the central database which includes all the tables in DB and the way they are interrelated) and we defined a number of functions in the fulfillment.

Line 480 - 482, Page 19, Score Analysis, Measures and Results

Fig (7a) displays the findings of the severity of depression within five groups - such as mild, moderate, no depression, severe and very severe - and the percentages which belong to each group of IDS-SR scoring system.

Line 493 - 495, Page 19, Score Analysis, Measures and Results

Fig (7b) displays the findings of the severity of depression within five groups - such as mild, moderate, no depression, severe and very severe - and the percentages which belong to each group of QIDS-SR scoring system.

Figure 8, An Overview on Scoring Systems (IDS-SR and QIDS-SR), Page 19, Score Analysis, Measures and Results

Line 497 - 498, Page 19, Score Analysis, Measures and Results

Fig 8 compares the 50 participants scored by the IDS-SR against those scored by the QIDS-SR, showing that the overall trend of these two scoring systems is identical.

Comment 10). The writer states that the participants are "keen on revealing their moods" several times throughout the paper. While it seems to be a reasonable assumption, there are no data or objective measures presented to support this claim.

Response 10). 

Thank you so much for the valuable comments and feedbacks. We have added the reference for the term “keen on revealing their moods” that claimed in this article. 

Summary of changes in the context are as follows:

Line 384 - 387, Page 15, Ethics and Consent Form, Methodologies

[42] Kim JY, Lee KM, Park SH. Evolution of revealing emotions. Physica A: Statistical Mechanics and its Applications. 2022;597:127268. doi: https://doi.org/10.1016/j.physa.2022.127268. 

[43] Xu L, Zhou X, Gadiraju U. Revealing the role of user moods in struggling search tasks. In: Proceedings of the 42nd International ACM SIGIR Conference on Research and Development in Information Retrieval; 2019. p. 1249–1252.

Comment 11). The writer refers to the study as a "non-clinical" study on several occasions. This term is confusing. I think what is meant here is that the sample is drawn from a non-clinical group. This point should be clarified as well.

Response 11). 

We appreciate the reviewer to pointing this out. This is a preliminary study which aims to provide a simple rather than a complex platform to validate its feasibility and efficacy. Therefore, we named this study as “non-clinical” trial. 

Summary of changes in the context are as follows:

Line 47 – 73, Page 3, Introduction

In summary, the community is currently open and positive on artificial intelligence and chatbots to help with mental health concerns. However, early detection before the onset of the disease is critical but is often overlooked. With this motivation, we proposed and developed a non-clinical chatbot, DEPRA, with social attributes to help users with mental health concerns and detect the illness in its early stage. This is a preliminary study which aims to provide a simple rather than a complex platform to validate its feasibility and efficacy.

DEPRA is capable of assisting participants examine their mental health conditions as a reference so that medical professionals can assist patients who are suffering from depression

• It was implemented following a conversation flow based structured early detection depression interview guide for the Hamilton Depression Scale (SIGH-D) [21] and Inventory of Depressive Symptomatology (IDS-C) [22] drawing on the experience of a group of psychiatrists. This makes the DEPRA chatbot a unique chatbot in that it shares the same set of clinically proved questions in the design of its questions for interaction with participants.}

• The validity of the questions used in the DEPRA chatbot was measured by applying closed group participation. This was arranged by designing open-ended questions for the participants. DEPRA is integrated with Facebook Messenger so, participants can interact with the chatbot through social media to share their responses.

• The questions asked by the DEPRA chatbot take approximately 30 minutes to complete. As the results suggest, the participants' responses indicate high satisfaction rates. They reported that the questions were easy to comprehend and respond to, it was not as time consuming as attending a face-to-face psychiatrist session, and they preferred the option to send text messages via social media platforms compared to talking to a psychiatrist in a consultation session.

Overall, the concept is very compelling, and I look forward to learning more about this area of study.

Response: We are grateful to the reviewer for these positive feedbacks.

Responses to Reviewer #2’s Comments

Overall, this is a strong piece of research with very interesting and timely findings! A few thoughts to strengthen this manuscript: 

We highly appreciate your positive feedbacks for this study.

Comment 1). A general proofread for grammar and readability issues is needed. For example, on page 2 in the Abstract Section, the sentence "First group comprises professional academic staff and HDR candidates, the second and third groups comprise relatives, friends, and volunteers who were recruited via social media promotions" could be more clearly stated. Further, on the same page another sentence begins: "computer-assisted mental health (CAMH)" without proper capitalization.

Response 1). 

We appreciated your constructive comments. We have completed second round of the proofreading by professional editor. Moreover, we have corrected all capitalization issues in this entire study. The readability and grammar issues should be resolved. 

Comment 2). Under the Related Works section and Psychological Intervention subsection, mention briefly any notable results of the studies you cite, as you do in the following Early Depression Detection Subsection. For example, were these projects effective?

Response 2). 

We appreciated your constructive feedback, and we agree. More analysis and comparison paragraph has been added into the Related Work section. 

Summary of changes in the context are as follows:

Line 78 – 80, Page 4, Related Works

We categorized the related works into two sections, namely works focus on technological-based depression detection and works which focus on chatbots which provide psychological interventions.

Line 82 – 102, Page 4, Psychological Intervention, Related Works

As the demand for medical professionals increases, utilizing chatbots to cooperate with and assist in the medical field will help with requests for assistance and ease the demand on medical practitioners. Ly et al. [24] conducted research to measure the effectiveness of smartphone apps in CBT interventions. A total of 28 participants (both males and females) took part in the Shim chatbot intervention which used a smart phone app as a text-only method of collecting data over a two-week period. The research confirmed that the participants' experiences and the output of their conversations with the Shim chatbot [25] can promote mental health. The duration of one face-to-face therapy session was 60 minutes. The results show that there is no significant inconsistency between a blended treatment (which comprised four face-to-face sessions and the smartphone application) and the full behavioural activation (which comprised including ten face-to-face sessions and no smartphone application) on the result variances. Both pre- and post-measurement and follow up actions were considered in this study. Sharma et al. [26] designed a chatbot application for Android devices which was designed as a virtual psychotherapist. Depression levels were measured on a scale of 0 to 4, with 0 being completely healthy and 4 being highly depressed. The strategy behind the question design was that for any optimistic response, the total score will increase by an x greater than zero value. The depression levels were divided into a) zero depression, absolutely healthy b) stressed c) highly stressed d) depressed e) highly depressed. The research results affirmed that it is quite difficult to extend therapy chatbot results because the assessments are not managed constantly.

Line 122 – 124, Page 4, Psychological Intervention, Related Works

The WorldBuilder chatbot has several limitations and the drawbacks of not being scalable and being overly reliant on the programmer. Unless this chatbot agrees to the existence of each object, there are no matches.

Line 129 – 137, Page 5, Psychological Intervention, Related Works

Woebot [34] is an automatic conversational chatbot based on CBT, implemented on Facebook Messenger. Experiments were conducted to evaluate Woebot using college students as participants. The results indicated that participants’ depressive symptoms dramatically decreased. The students commented that their interactions with Woebot were more enjoyable than the therapy sessions held by health care professionals. However, Woebot does not provide a comprehensive process of CBT and it mainly deals with psychoeducation for stress control. Participants need maintain their level of enthusiasm to interact with Woebot for a period of time before the final results are achieved

Line 138 – 150, Page 4, Psychological Intervention, Related Works

In an unblinded trial conducted by Fitzpatrick et al. [36], 70 participants from a university community social media site in the age range of 18 to 28 years old were recruited online to evaluate the acceptance rate of college students who reported feelings of anxiety and depression to engage with a conversational agent and participate in a self-help program. The average age of the participants was 22.2 years old, 47 out of 70 were female, 54 out of 58 were non-Hispanic and 46 out of 58 were Caucasian. The participants were randomized to receive either 2 weeks or up to 20 sessions of self-help content derived from CBT principals in a conversational format with the text-based conversational agent (Woebot) (n=34) or were directed to the National Institute of Mental Health (NIMH) e-book, “Depression in College Students” as an informative control group (n=36). At the next stage, all the participants were encouraged to complete the web-based versions of PHQ-9, GAD-7 and Positive and Negative Affect Scale at baseline and two-to-three weeks later.

Line 199 – 227, Page 6, Psychological Intervention, Related Works

Fulmer et al [20] proposed Tess which the interactions generated by about 354 participants with Tess depressions modules were considered to understand chatbot usage within the modules. In the research, the researchers conducted a randomized controlled trial to assess impact of using an integrative therapeutical AI agent, Tess, a mental health chatbot that provides self-help chats through text messages and evaluated its efficacy in reducing self-identified symptoms of depression and anxiety in college students. The study participants engaged in natural conversations with Tess via Facebook Messenger. No demographic variables of the participants were collected, so no data for this part of the experiment is available. Tess is able to react to the participants' emotional needs by analysing the conversational content either through textual interaction or by Facebook Messenger sampling. Tess is designed to play the role of a traditional therapist; however, it is not a replacement for face-to-face therapy by a professional. The experiment identified several limitations in the use of Tess. First, it is not possible to determine why a user stops interacting with Tess, for example, were they redirected to a different module, or did they simply lose interest in continuing with the interactions etc. Also, of the 354 participants, two claimed there were technical issues in the design of Tess and they stated that they could not finalize their interaction within the experiment and continue the sessions due yo these issues.

Our proposed DEPRA chatbot falls under this bot family category as a Depression Detection Bot, the aim of which is to circumvent some of the existing challenges that have been discussed in this section. The main difference between this research and the related works conducted so far is that the DEPRA chatbot gathers data based on a series of psychologically approved questions and its purpose is to triage and detect the early signs of depression. However, the available chatbots mostly focus on the depression disorder itself and try to find a better cure. In other words, the DEPRA chatbot aims to assist medical professionals detect and cure depression in its early stage and it does not aim to replace a medical professional. DEPRA was implemented to further investigate the application of AI to interact with users, collect their responses to questions on depression symptoms and identify depression and estimate the level of its severity.

Comment 3). The Participant Recruitment on page 8 needs clarification. For example, after reading "We used three channels for recruitment (Fig 6) - the first group was university students and university professional staff either full-time, part-time or causal" and looking at Figure 6, it is still unclear how participants were recruited into this study at various points.

Response 3). 

Thanks for your point to make this section more comprehensive. We have redrawn the figure to improve the readability as well as restructured the sentence within the paragraph. 

Summary of changes in the context are as follows:

Line 350 – 367, Page 14, Participant Recruitment

The participant recruitment is based on three categories: a) Victoria University staff and HDR Students; b) friends and family members (non-organic group); c) Facebook page (organic group). We used three channels for recruitment (Fig~\\ref{fig6}) - the first group was university students and university professional staff either full-time, part-time or causal. We emailed a broad group of potential participants using three email accounts with the Webhook link to invite them to participate in this research. We required the academic group to assist in two tasks. Firstly, we recruited 10 participants (N=10) to share their ideas and responses in a closed group. This was crucial to designing the utterances on Dialogflow as the platform for the DEPRA chatbot. Secondly, we asked the same group to participate in data collection to assist with the research. We then emailed another 65 university staff and HDR students to invite them to participate. We received 16 complete records in our AWS central database. Current academics and HDR candidates were the point of contact as they were presented with the details of the research in weekly and monthly meetings and with these series of updates, they played a critical role in the understanding and comprehension of the requirements of the research. As a consequence, they assisted in responding to the set of questions and helped to generate the potential responses required for the design of the chatbot.

Fig 6. Participant Recuritment Flow – 3 Groups Indicating Statistics of Complete Response and Dropout Rate, Page 15, Participant Recruitment, Methodologies

Comment 4). Further, in your abstract you mention that participants "included academics and HDR candidates who are conscious, vigilant and have a clear understanding of the questions." Describe how this was assessed in the Participant Recruitment section.

Response 4). 

We appreciate your comments on the specific section of the abstract. We have revised the paper to have better descriptions on Participant Recruitment. 

Summary of changes in the context are as follows:

Line 362 – 367, Page 14, Participant Recruitment

Current academics and HDR candidates were the point of contact as they were presented with the details of the research in weekly and monthly meetings and with these series of updates, they played a critical role in the understanding and comprehension of the requirements of the research. As a consequence, they assisted in responding to the set of questions and helped to generate the potential responses required for the design of the chatbot.

Comment 5). Under the Limitations section on page 14, it would be helpful if you could speak to any potential bias relating to recruitment of the research team's friends and family as participants for this study, as compared to the general Australian public.

Response 5). 

We appreciate that sharing your opinion on this point. In this revised version, we have added more discussions and future plans to further address the concerns of recruitment and statical analysis. 

Summary of changes in the context are as follows:

Line 643 – 658, Page 23, Discussion and Future Works, Planning of Validation

As a non-clinical trial study, this study also aims to identify whether DEPRA has the greatest probability of success, assess its safety, and build solid scientific foundations before transitioning to a future clinical development phase. Thus, this study created a realistic experimental environment and recruited local users to better validate the usability of DEPRA. The results of the experiment demonstrate that users were relatively more receptive to DEPRA and accepted the effectiveness of DEPRA for the early detection of mental health concerns. For the scientific validation of the questionnaire, we approached several psychologists and a team of experts in related professions to undertake an authoritative validation. In addition, the questionnaire will be improved according to professional guidance and will be reflected in the forthcoming works. Moreover, we are currently using the IDS-C and QIDS-C to validate the results. The work of the comparison group will continue to be developed, including and not limited to the automated scoring system, the comparison of the scoring system with the expert team's scoring results, and the comparison of the scoring results with the user's perception.

Line 567 – 578, Page 22, Discussion and Future Works, Participant Recruitment

To comply with the Victoria University ethics guidelines, we were only permitted to invite participants who were living in Australia and aged between 18 to 80 years old in this non-clinical trial. However, we believe this work will have a greater impact and will receive more widespread support if this article is published. Moreover, the discussion with the university has started and is scheduled for further development. The future target audience will vary in terms of their country of origin, ethnicity, age, and gender and will include those who potentially have mental health concerns.

Line 576 – 599, Page 22, Discussion and Future Works, Participant Dropout Rates

This study comprises three participation groups, namely Group 1: staff and HDR students, Group 2: friends who interact via Facebook, and Group 3: the open access organic group. The total dropout rate for each group from first contact was 79%, 48%, and 94%, respectively. The third group has the highest rate of dropout, which is to be expected. People who were curious found the DEPRA chatbot on the Facebook page on their own and decided to interact. However, many (91%) were unwilling to continue after reading the consent form. Moreover, we can make reasonable assumptions after a few informal interviews that the participants were worried about revealing their personal information when they required to sign the consent form, especially when the project did not have a solid manifestation, such as scientific articles that have already been published. The consent form dropout rate for Group 2 was the highest at 15% due to the fact that the authors had the opportunity to explain the project in detail.

Another interesting finding is the dropout rate before the consent form was sent, where Group 3 had lowest dropout rate at 18% and Group 1 had highest at 65%. 70 of the 85 invited participants reached the point of reading and signing the consent form. These results demonstrate that the depression detection chatbot has significant potential on the social media platform. A study by Google [45] also reported that the search interest for “therapists near me” hit record highs in 2021, and the phrase "why do I feel anxious for no reason" also hit an all-time high this year, spiking at more than 400%

In summary, the characteristics of DEPRA make it an attractive platform on social media to help users who are concerned about their depression level. Furthermore, we believe this publication will increase the confidence level of volunteers to contribute to future studies.

Line 601 – 632, Page 23, Discussion and Future Works, Participant Satisfaction Scales

We received 29 valid survey submissions which included responses to questions regarding the ease of answering questions in the questionnaire, the acceptance of the human-computer interaction, the interaction time, the question sequence in terms of depression severity level guidance, stress detection, and the liklihood that they will recommend the use of DEPRA to friends and family. It is noteworthy that 94\\% of all questionnaire responses were satisfactory in relation to DEPRA. This indicates that the majority of users believe that DEPRA is useful in relation to their depression detection and they are satisfied with the interaction. Although the number of participants was limited by geographic location and interpersonal relationships, a certain percentage of the total number of users was anonymous. Therefore, this satisfaction level is still informative in this preliminary work. In future work, we plan to involve participants from a wider range of ethnicities in the experiment to obtain more comprehensive results.}

After we analyzed the results of each questionnaire, we made a very explorable observation. In Q3, we measured the users’ perceptions of text messages and noted that three users returned negative responses. This question received the largest negative score of all the questions in the questionnaire. From this result, we speculate that text input becomes a relatively unreasonable request when the information described by the user involves a long paragraph. This is understandable as inputing lengthy text leads to problems such as mis-touching the send button, the input box not being able to display all the content of a lengthy text, the long time required to type a lengthy text causing the page response time to expire, etc. Therefore, we will use this user feedback as a reference to explore and study natural language processing in voice inputs and its processing in our future works.

Furthermore, Q2 explored whether the interaction with DEPRA was satisfactory to users in terms of the time spent compared with attending the psychiatrist session. This is the only question for which we received a score of 1/5. Possible reasons for this result may be the number of questions, whether the content is easy to understand for non-first language English speakers, whether the questions are too long, or whether the user needs to provide more feedback for some questions, etc. This questionnaire will also provide a very high reference value for DEPRA in its future work.

Comment 6). In your introduction you posit the following hypothesis: "We predict that the absence of a booking requirement and the flexibility of participating on their own schedule will eliminate an impediment to obtaining preparatory help and avoiding social embarrassment." Please note how that factors in with the overall conclusions of this study on page 15?

Response 6). 

Thanks for your attention to details and mentioning the point. We added more discussions in satisfaction scales which including the results of related questioners. 

Summary of changes in the context are as follows:

Line 601 – 632, Page 23, Discussion and Future Works, Participant Satisfaction Scales

We received 29 valid survey submissions which included responses to questions regarding the ease of answering questions in the questionnaire, the acceptance of the human-computer interaction, the interaction time, the question sequence in terms of depression severity level guidance, stress detection, and the liklihood that they will recommend the use of DEPRA to friends and family. It is noteworthy that 94\\% of all questionnaire responses were satisfactory in relation to DEPRA. This indicates that the majority of users believe that DEPRA is useful in relation to their depression detection and they are satisfied with the interaction. Although the number of participants was limited by geographic location and interpersonal relationships, a certain percentage of the total number of users was anonymous. Therefore, this satisfaction level is still informative in this preliminary work. In future work, we plan to involve participants from a wider range of ethnicities in the experiment to obtain more comprehensive results.}

After we analyzed the results of each questionnaire, we made a very explorable observation. In Q3, we measured the users’ perceptions of text messages and noted that three users returned negative responses. This question received the largest negative score of all the questions in the questionnaire. From this result, we speculate that text input becomes a relatively unreasonable request when the information described by the user involves a long paragraph. This is understandable as inputing lengthy text leads to problems such as mis-touching the send button, the input box not being able to display all the content of a lengthy text, the long time required to type a lengthy text causing the page response time to expire, etc. Therefore, we will use this user feedback as a reference to explore and study natural language processing in voice inputs and its processing in our future works.

Furthermore, Q2 explored whether the interaction with DEPRA was satisfactory to users in terms of the time spent compared with attending the psychiatrist session. This is the only question for which we received a score of 1/5. Possible reasons for this result may be the number of questions, whether the content is easy to understand for non-first language English speakers, whether the questions are too long, or whether the user needs to provide more feedback for some questions, etc. This questionnaire will also provide a very high reference value for DEPRA in its future work.

---

## [Decision Letter · Decision Letter 1]

12 Jul 2022

PONE-D-22-04147R1

Early Detection of Depression Using a Conversational AI Bot: a Non-Clinical Trial

PLOS ONE

Dear Dr. Ahmed,

Thank you for submitting your manuscript to PLOS ONE. After careful consideration, we have decided that your manuscript does not meet our criteria for publication and must therefore be rejected.

I am sorry that we cannot be more positive on this occasion, but hope that you appreciate the reasons for this decision.

Kind regards,

Sriparna Saha, PhD

Academic Editor

PLOS ONE

Reviewers' comments:

Reviewer's Responses to Questions

**Comments to the Author**

1. If the authors have adequately addressed your comments raised in a previous round of review and you feel that this manuscript is now acceptable for publication, you may indicate that here to bypass the “Comments to the Author” section, enter your conflict of interest statement in the “Confidential to Editor” section, and submit your "Accept" recommendation.

Reviewer #1: (No Response)

Reviewer #2: All comments have been addressed

2. Is the manuscript technically sound, and do the data support the conclusions?

Reviewer #1: Partly

Reviewer #2: Partly

3. Has the statistical analysis been performed appropriately and rigorously? 

Reviewer #1: I Don't Know

Reviewer #2: Yes

4. Have the authors made all data underlying the findings in their manuscript fully available?

Reviewer #1: Yes

Reviewer #2: Yes

5. Is the manuscript presented in an intelligible fashion and written in standard English?

Reviewer #1: No

Reviewer #2: Yes

6. Review Comments to the Author

Reviewer #1: While the authors have done work to improve the readability of the paper, the overall structure remains difficult to follow and confusing to the extent that it is challenging to understand the aims and phases of the trial. At this time, it seems that a complete rewrite would be necessary to make this manuscript appropriate for publication.

Reviewer #2: The authors took great care to use the reviewer feedback and strengthen this manuscript. And I must reiterate that the data presented here are timely and has important implications for our field. Some comments to strengthen this piece further.

1. This manuscript would benefit from an additional review for grammar and readability issues. Some of the sentences as written are written in a manner that is unclear and takes several repeated reads for comprehension. Some examples of this:

The sentence on line 177 that reads "Philip et al. [?] conducted research with 179 participants for a period of 1 day using an of Embodied Conversational Agent (ECA) which is a form of intelligent user interface that interacts with the environment through a physical body" is difficult to understand (partially due to the extra "of" in the sentence) and does not go on to explain what the "physical body" is referring to.

Additionally, within this paragraph, the manner in which the Phillip study is described needs clarification. Line 185 reads "A total of 35 participants (19.6 percent) were diagnosed with Major Depressive Disorder (MDD) confirmed by a psychiatrist." Then line 193 reads "The ECA interview revealed the following statistics: 58 percent had mild depression, 77 percent had moderate depression and 84 percent had severe depression." Does this mean that the ECA interview is over-diagnosing depression, as they assessed a greater proportion of the sample as having depression than the psychiatrist did? This would point to limited utility of the ECA interview.

Another example - The sentence on line 199 that reads "Fulmer et al. [?] proposed Tess which the interactions generated by about 354 participants with Tess depressions modules were considered to understand chatbot usage within the modules" is confusing without first understanding what Tess actually is. Since the following sentence that that reads "the researchers conducted a randomized controlled trial to assess impact of using an integrative therapeutical AI agent, Tess, a mental health chatbot that provides self-help chats through text messages" actually explains what Tess is, this sentence should have been presented before describing the interactions of the 354 participants with Tess.

An outside reader less familiar with the work of the study would likely be able to provide additional edits to increase the readability of this piece.

2. In the abstract the authors state "As an integrative psychological AI, DEPRA, emerges as a feasible option for delivering emotional and physical support to participants with early depression detection capabilities" in the Conclusion. While this paper clearly presents data that support the feasibility of DEPRA in early detection of depression, it does not provide data on how the chatbox provides "emotional or physical support." It is not clear how the chatbox asking questions surrounding the presence of depressive symptoms provides emotional support to the users, and it is not enough to say that users rated the experience of using the chatbox as satisfactory to support this conclusion. For example, did the chatbox also make validating, empathic, or encouraging statements after the users provided answers describing their symptoms?

Overall, this revision has adequately addressed the areas of improvement from the original version. Some additional clarification overall would greatly strengthen this piece!

7. PLOS authors have the option to publish the peer review history of their article (what does this mean?). If published, this will include your full peer review and any attached files.

Reviewer #1: No

Reviewer #2: No

- - - - -

---

## [Editor Report · Decision Letter 2]

8 Nov 2022

PONE-D-22-04147R2

Early Detection of Depression Using a Conversational AI Bot: a Non-Clinical Trial

PLOS ONE

Dear Dr. Ahmed,

Thank you for submitting your manuscript to PLOS ONE. After careful consideration, we feel that it has merit but does not fully meet PLOS ONE’s publication criteria as it currently stands. Therefore, we invite you to submit a revised version of the manuscript that addresses the points raised during the review process.

ACADEMIC EDITOR: I have one concern regarding the paper. The ethical considerations are not discussed by the authors in the manuscript. Have they followed proper ethical guidelines for conducting this research?

We look forward to receiving your revised manuscript.

Kind regards,

Sriparna Saha, PhD

Academic Editor

PLOS ONE
---

## [Author Response · Author response to Decision Letter 2]

24 Nov 2022

Comment 1) I have one concern regarding the paper. The ethical considerations are not discussed by the authors in the manuscript. Have they followed proper ethical guidelines for conducting this research?

Response: 

The manuscript discusses the ethical consideration in detail on page#13 as a separate sub-section. The authors also included the approved complete ethics application and consent form in the Appendix.

For your ready reference, here is the excerpt or summary from the section

This research is registered under ethics approval number HRE 20-184 at Victoria University, Melbourne, Australia. It involves a non-clinical population of participants who reside in Australia at the time of the research. The participants are between 18 and 80 years old and willing to reveal information on their moods [42] and emotions [43] in a scientific trial. Appendix C provides the consent form for the participants involved in the research, and Appendix D provides the Ethics Application related to this study. 

The participants indicated their willingness to interact with the DEPRA chatbot and participate in the study by signing the consent form electronically. The participants were prompted to access the consent form after contacting DEPRA. All participants must agree to the terms and conditions in the consent form, which notes that DEPRA may record and analyse their responses to the questions relating to moods, sleep habits, appetite, general health symptoms, etc. 

This research deals with human beings and their moods and behaviours in general. The ethics committee follows strict guidelines and conditions to confirm the validity of the methods and approach used by the research team to conduct the research. One of the ethics committee's main concerns is how the study maintains the participants' privacy. After completion of the data collection by the research team, the data is stored on an R drive and a computer with limited access. Only the research team members are allowed access to the raw data. The participants are assured that no unauthorised users will access the raw data. All data were de-identified to protect the privacy of the participants. 

Besides this dedicated section, the authors also discuss the ethical considerations on page#12 under the subsection 'Participant Recruitment' and page#18~19 under 'Discussions and Future Work.'

---

## [Editor Report · Decision Letter 3]

15 Dec 2022

Early Detection of Depression Using a Conversational AI Bot: a Non-Clinical Trial

PONE-D-22-04147R3

Dear Dr. Ahmed,

We’re pleased to inform you that your manuscript has been judged scientifically suitable for publication and will be formally accepted for publication once it meets all outstanding technical requirements.

Kind regards,

Sriparna Saha, PhD

Academic Editor

PLOS ONE
---

## [Editor Report · Acceptance letter]

24 Jan 2023

PONE-D-22-04147R3 

Early Detection of Depression Using a Conversational AI Bot: a Non-Clinical Trial 

Dear Dr. Ahmed:

I'm pleased to inform you that your manuscript has been deemed suitable for publication in PLOS ONE. Congratulations! Your manuscript is now with our production department. 

Kind regards, 

on behalf of

Dr. Sriparna Saha 

Academic Editor

PLOS ONE